# Intermolecular Coulombic decay in liquid water competes with proton transfer and non-adiabatic relaxation

Pengju Zhang [1,2] ✉, Joel Trester [2], Jakub Dubský[3], Přemysl Kolorenč [4], Petr Slavíček [3] ✉ & Hans Jakob Wörner [2] ✉

Despite decades of research, our understanding of radiation damage in aqueous systems remains limited. The recent discovery of Intermolecular Coulombic Decay (ICD) following inner-valence ionization of liquid water raises interesting questions about its efficiency as a major source of low-energy electrons responsible for radiation damage. To investigate, we performed electron-electron coincidence measurements on liquid $H_2O$ and $D_2O$ using a monochromatized high-harmonic-generation light source, detecting ICD electrons in coincidence with photoelectrons from the $2a_1$ shell. We find that the ICD efficiency $\gamma$ is below unity in both liquids and that $\gamma(H_2O)/\gamma(D_2O) = 0.86 \pm 0.03$. Ab initio calculations reveal that ICD competes with proton transfer and non-adiabatic relaxation, which can close the ICD channel. A multi-scale stochastic model incorporating solvent effects reproduces these efficiencies. Our combined experimental and theoretical results suggest that the higher ICD efficiency in $D_2O$ arises from slower proton transfer and non-adiabatic transitions, highlighting the crucial role of nuclear motion in liquid-phase ICD and advancing the understanding of radiation damage.

Non-local electronic relaxation processes turned out to be important in understanding the complex energy-transfer dynamics in condensed matter exposed to ionizing radiation. Among these processes, Interatomic or Intermolecular Coulombic Decay (ICD) has been of broad interest due to its substantial ramifications in photochemistry and radiation chemistry[1]. Following the removal of an inner-valence or core electron, an outer-valence electron from the same atom or molecule can fill the vacancy, and transfer the excess energy to a neighboring entity that gets ionized in the process ($H_2O^{*+} \cdots H_2O \rightarrow 2\,H_2O^+ + e^-$). The potential significance of ICD in radiation chemistry has been suggested by several studies, which demonstrated that low-energy electrons ($E_{kin} < 15\,eV$), such as those created during ICD, can cause DNA strand breaks in living tissues[2], and the radical species can react with surrounding biomolecules[3,4], causing further damage to the biologically relevant systems. The non-local electronic decays can appear in a cascade, creating hotspots of radiation damage near the originally ionized centers[5,6].

Since the pioneering theoretical prediction by Cederbaum et al.[7], the existence of ICD has been extensively identified in numerous systems, including van-der-Waals clusters[8–17], molecular anions[18], nanodroplets[19–21], hydrogen-bonded clusters[22–26], solids[27], solvated ions[28] and unbound molecules[29]. For reviews of this topic see refs. [1,30].

The first observation of inner-valence ICD in liquid water[31] was made possible by the development of electron-electron-coincidence spectroscopy with liquid microjets. This advance now allows us to raise the crucial question of the mechanism of ICD in the liquid phase and to

[1]Beijing National Laboratory for Condensed Matter Physics and Institute of Physics, Chinese Academy of Sciences, Beijing, China. [2]Laboratory of Physical Chemistry, ETH Zürich, Zurich, Switzerland. [3]Department of Physical Chemistry, University of Chemistry and Technology, Prague, Czech Republic. [4]Charles University, Faculty of Mathematics and Physics, Institute of Theoretical Physics, Prague, Czech Republic. ✉e-mail: pengju.zhang@iphy.ac.cn; petr.slavicek@vscht.cz; hwoerner@ethz.ch

understand which processes possibly compete with it. ICD usually has an intrinsic lifetime on the order of tens to hundreds of femtoseconds in small rare-gas clusters[14,32,33], which leads to its efficiency being close to 100% whenever the local Auger process is energetically closed, thereby quenching all other relaxation processes.

Compared to van-der-Waals clusters, the situation is richer in hydrogen-bonded systems, where proton transfer competes with the electronic autoionization pathways. In the case of inner-valence ionized water clusters, the ICD efficiency was measured to lie far below unity[24]. Both the low efficiency and an observed isotope effect were assigned to the suppression of ICD by the competing inter-molecular proton transfer. However, although the extrapolation of the size-dependent efficiency suggested a value below one for the liquid bulk, the accompanying calculations indicated that the ICD channel in liquid water was not suppressed by proton transfer[24].

In the present work, we resolve this controversy and explore the coupling of electron and nuclear dynamics in liquid water by a com-bination of coincidence measurements and ab-initio molecular dynamics and stochastic simulations including the solvent effects. On the experimental side, we demonstrate the occurrence of ICD in both

regular and deuterated bulk liquid water. In addition, the relative ICD efficiency between regular and deuterated water is determined by quantitative analysis of the ICD+photoelectron coincidence spectrum. The isotope effect allows for clarifying the role of nuclear dynamics evolving in the electronic relaxation process. The measured data are interpreted with a multi-scale simulation model. The molecular dynamics simulations confirm the competition between proton transfer and ICD initiated by the ionization of the $2a_1$ inner-valence band in liquid water. We explicitly consider the solvent effect by including up to several hundreds of water molecules in the simulation, which identifies the convergence between large-size water clusters and liquid water for these two competitive pathways. We then formulate a stochastic model that allows us to model all measurable quantities - distribution of electron kinetic energies, ICD efficiencies, and isotope effects. The simulations provide a sharp view on the competing electron-nuclear processes following the inner valence ionization.

## Results

Figure 1 shows the electron-electron-coincidence spectra of electron pairs using an XUV photon energy of 63.5 eV (H41 of 800 nm). Panel (b)

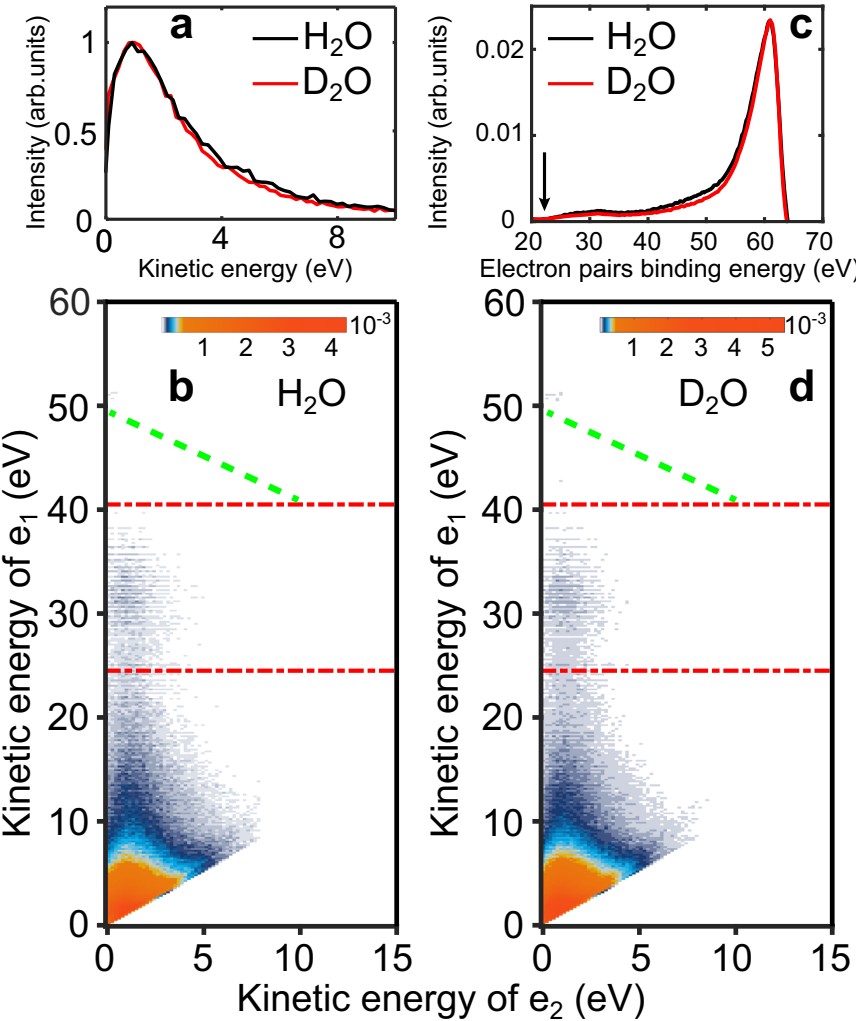

**Fig. 1 | Coincidence measurements of ICD in liquid $H_2O$ and $D_2O$ with XUV photons of 63.5 eV. a** Comparison of the energy spectra of ICD electrons from liquid $H_2O$ (black solid) and $D_2O$ (red solid). The energy spectra are obtained by integrating over the rectangle area delimited by dashed red lines ($e_1 \in [24.5, 40.5]$ eV) from (**b, d**), respectively. The spectra are normalized to their maximum for comparison. **c** Comparison of the electron-pair spectra of liquid $H_2O$ (black solid) and $D_2O$ (red solid) as a function of their electron-pair binding energy, obtained by

summing along lines of constant total energy ($E(e_1) + E(e_2)$), i.e. lines parallel to the green dashed lines in (**b, d**), respectively. The spectra are normalized to the total electron-pair counts for comparison. **b, d** Coincidence map of electron pairs pro-duced by ionization of the inner-valence $2a_1$ band in liquid $H_2O$ and $D_2O$ (note the logarithmic intensity scale), respectively. The area between the two red lines is dominated by $2a_1$-photoelectron/ICD-electron pairs.

shows the coincidence map of liquid $H_2O$, where the counts of electron pairs with respect to a given kinetic energy of the fast electron ($e_1$) as well as an energy of the slow electron ($e_2$) are displayed on the vertical and horizontal axis, respectively. Apart from the dominant emission of very slow (<5 eV) electron pairs, which consist of a secondary electron originating from the inelastic scattering or/and electron-impact ionization process in the bulk and the primary electron that has lost energy, a clear island consisting of a fast electron centered around $e_1 \approx 32$ eV and a slow electron extended above 5 eV is observed. These events are the electron pairs with photoelectrons from the $2a_1$ inner-valence band and the slow ICD-electrons. Panel (d) is the same as panel (b) but for liquid $D_2O$. Overall, the character of the $D_2O$ result is very similar to that of $H_2O$, and the island of ICD+photoelectron pairs is also observed. Nevertheless, the secondary-primary electron pairs are even more pronounced (see Fig. S2 in Supplementary Information), which is caused by a higher electron-impact ionization cross section of $D_2O$[34]. The doubly charged final states, which are populated via either direct photo-double-ionization, single photoionization of outer-valence band followed by electron-impact ionization or the ICD process, can be obtained by integrating the signals in the coincidence map along lines of constant total energy ($E(e_1) + E(e_2)$), i.e. along diagonal lines parallel to the dashed green line in panels (b) and (d). The comparison of the electron-pair spectra of liquid $H_2O$ (black solid) and $D_2O$ (red solid) is shown in panel (c). The spectra are normalized to the total electron-pair counts for comparison. The onset (black arrow) of this distribution indicates the minimum two-hole-state energy expected, which corresponds to twice the ionization potential of the highest occupied molecular band ($1b_1$) of liquid water[35]. The pronounced distribution ranging from 23 eV to 38 eV is attributed to the creation of delocalized pairs of outer-valence ($1b_1$, $3a_1$ or $1b_2$) vacancies due to ICD. A small fraction of the electron pairs in the higher-binding-energy region (≳35 eV) is due to the creation of local double-hole states, whereas the highest-binding energy part is dominated by secondary electrons created through electron-impact ionization of liquid water by primary photoelectrons[31,36].

The spectra of ICD electrons are obtained by summing the signals in the coincidence map between the two red dashed-dotted lines in panel (b, d) for liquid $H_2O$ (black solid) and $D_2O$ (red solid), shown in panel (a). The spectrum exhibits a quasi-exponential decay profile, in particular for kinetic energies above 2.5 eV, and is consistent with the theoretical predictions for small water clusters[37] and their experimental measurements[22–24]. For the two liquids, the ICD spectra display a peak around 1 eV, which is probably caused by the presence of an escape barrier on the order of 0.2 eV[36].

Having verified the occurrence of ICD in liquid water, we now perform a more detailed analysis of its efficiency in both $H_2O$ and $D_2O$. From the measured 2D electron-electron coincidence data (see Fig. 1(b, d)), in principle, we can extract the absolute ICD efficiency $\xi_{ICD}$, defined as the fraction of $2a_1$ vacancies that relax via ICD. Formally, $\xi_{ICD}$ is described by the ratio of coincident electron pairs $P(E_{2a_1}, E_{ICD})$ to the number of $2a_1$ photoelectrons, $P(E_{2a_1})$, where $E_{2a_1}$, $E_{ICD}$ represent the kinetic energies of the fast photoelectron and the slow ICD electron, respectively. Considering the actual experimental conditions (see section 2 in the supplementary information (SI)), we obtained the ICD efficiency in liquid water as:

$$\xi_{ICD} = \frac{P(E_{2a_1}, E_{ICD})}{P(E_{2a_1})} \cdot \frac{1}{\gamma(E_{ICD})} \qquad (1)$$

where $\gamma(E_{ICD})$ is the experimental detection efficiency of the ICD electron. Therefore, to know the absolute ICD efficiency, one has to determine the absolute instrumental detection efficiency of the ICD electrons. As shown in Fig. 1(a), the ICD-electron spectrum extends from 0 to 10 eV. The detection efficiency of the magnetic bottle time-of-flight spectrometer is less than 100%. Furthermore, the

detection efficiency is very sensitive to the specific configuration of the magnetic fields induced by the permanent magnet together with the solenoid coil in the flight tube. Absolute efficiencies can only be achieved when the absolute collection efficiencies are known, which is particularly challenging for slow electrons. Moreover, the spectrometer has to maintain an identical configuration during the long-term measurements necessary to acquire the coincidence maps. These complexities prevent the extraction of the absolute ICD efficiency of liquid water. Nevertheless, since the two samples are measured under identical experimental settings (see experimental methods) and their ICD energy distributions are distinguishable (see Fig. 1a, the relative ICD efficiency between liquid $H_2O$ and $D_2O$ can be directly obtained.

Figure 2 (e) shows the ICD efficiency after $2a_1$ ionization with different XUV photon energies. Panel (a) represents the photoelectron energy spectrum $P(E_{2a_1})$ of liquid $H_2O$, obtained by the summation of the sorted single-, double-, triple- and quadruple-hit events, i.e. total $2a_1$ photoionization events. Panel (b) corresponds to the coincident ICD + photoelectron pair spectrum $P(E_{2a_1}, E_{ICD})$ of liquid $H_2O$, integrated over the entire ICD-electron energy distribution, i.e., 0–10 eV. Both spectra are derived from the same data set shown in Fig. 1. A linear background originating from the inelastic scattering as well as direct and indirect photo-double-ionization are subtracted from the raw data (see section 2 in SI). The same spectra for liquid $D_2O$ are shown in panels (c) and (d) respectively.

Panel (e) displays the relative ICD efficiencies between liquid $H_2O$ and $D_2O$ obtained by two different background subtraction methods. For comparison, we also show the ICD efficiencies of $H_2O$ (black) and $D_2O$ (green) as well as their ratio (blue) for a cluster size of $N \sim 72$, extracted from ref. 24, in Fig. 2(e). Similar to the water cluster, a clear isotope effect is observed, with liquid $D_2O$ exhibiting a higher ICD efficiency compared to liquid $H_2O$. This suggests a competition between the ICD process and other relaxation channels. Additionally, the relative ICD efficiency remains approximately constant over a broad range of XUV photon energies, within the experimental uncertainties. Interestingly, the relative ICD efficiency of liquid $H_2O$ vs. $D_2O$ is consistent with that of normal vs. deuterated water clusters with sizes above 60 monomers. Furthermore, the cluster-size-dependent ICD efficiency reported in ref. 24 shows that the absolute ICD efficiency increases with cluster size but always remains below unity. In analogy to the cluster results, our liquid-phase results indicate that a competing relaxation process, such as proton transfer is still active in the liquid-phase environment.

The traditional perspective of ICD in water relies on the single-electron picture: the electron in the $2a_1$ state is ejected and the corresponding hole is formed. We discuss below that this picture is oversimplified as the $2a_1$ state is actually embedded in a large number of satellite states that are heavily populated during the ionization process. Nevertheless, focusing on the $2a_1$-one-hole state is a good starting point for gaining insight into the interplay between the electronic and nuclear dynamics.

The simplified scheme of this complex process is outlined in Fig. 3. Upon the removal of a $2a_1$ electron, the ICD channel is energetically open. The characteristic timescale of the electronic decay was previously estimated as tens of femtoseconds[24]. This is a sufficiently long time for the fastest nuclear dynamics to take place. For example, the proton transfer for water ionized in the outer shells is faster than 50 fs[38]. The proton transfer in the core-ionized state is even faster, taking place in about 10 fs[39]. Ultrafast proton transfer was also identified in the $2a_1$ one-hole state in a water dimer[24]. The $2a_1$ energy then decreases faster than the energies of the doubly-ionized state, and, at a certain point, the ICD decay channel energetically closes.

In reality, the situation is further complicated by the so-called breakdown of the molecular-orbital picture[40,41]. The strong correlation in the molecular inner-valence spectral region manifests itself through

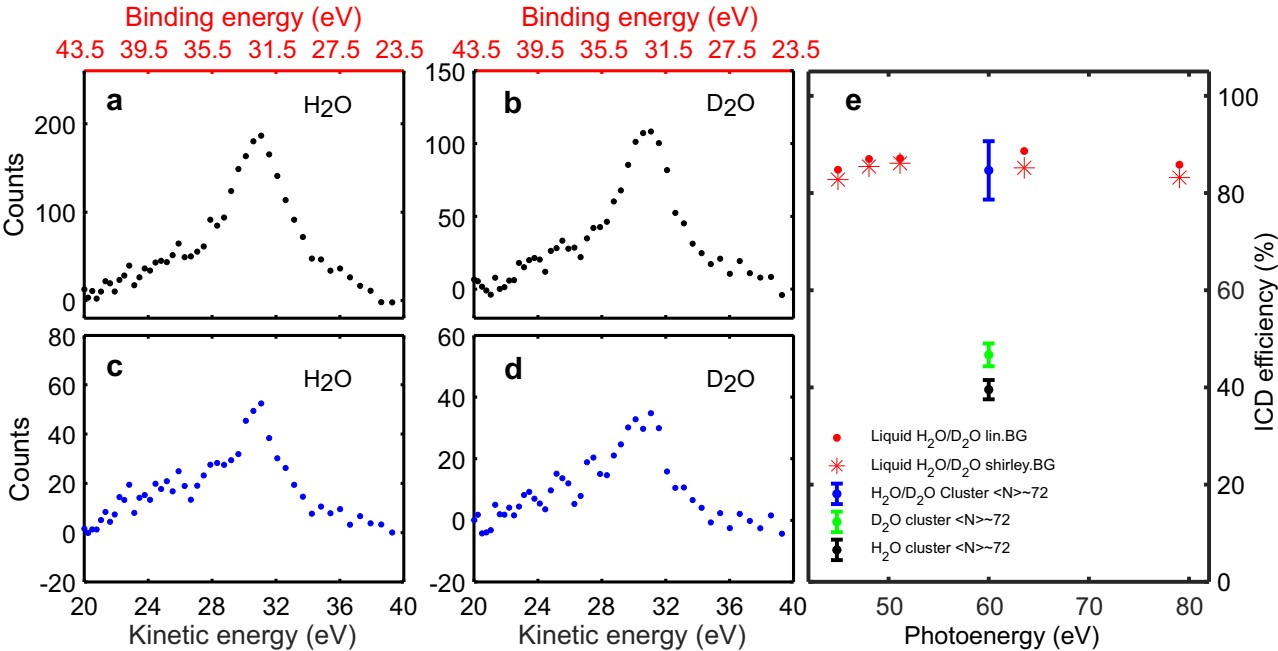

**Fig. 2 | Efficiency of ICD after photoionization of the 2a₁ band in liquid water with different XUV photon energies. a** Photoelectron energy spectrum of liquid $H_2O$ in the region of $2a_1$ binding energies. **b** Energy spectrum of ICD+photoelectron pairs obtained from the liquid $H_2O$ coincidence map in the region of $2a_1$ binding energies. **a**, **c** are derived from the same data set shown in Fig. 1. Here, a linear background was subtracted. Electron energy is represented as kinetic energy (bottom axis) or binding energy (top axis). **b**, **d** Same as (**a**, **c**) but for $D_2O$.

**e** Relative ICD efficiencies of $H_2O$ and $D_2O$ obtained by two different background subtraction methods (see section 2 in SI): a simple linear background (lin.BG) model, consistent with the analysis in ref. 55 and a Shirley background (shirley.BG) model, which is widely used in x-ray photoelectron spectroscopy analysis[56]. The absolute ICD efficiencies of $H_2O$ and $D_2O$ clusters ($\langle N \rangle$ - 72) measured at a photon energy of $h\nu$ = 60 eV are shown for comparison[24]. Error bars are adapted from ref. 24, which represent the standard deviation due to stochastic errors.

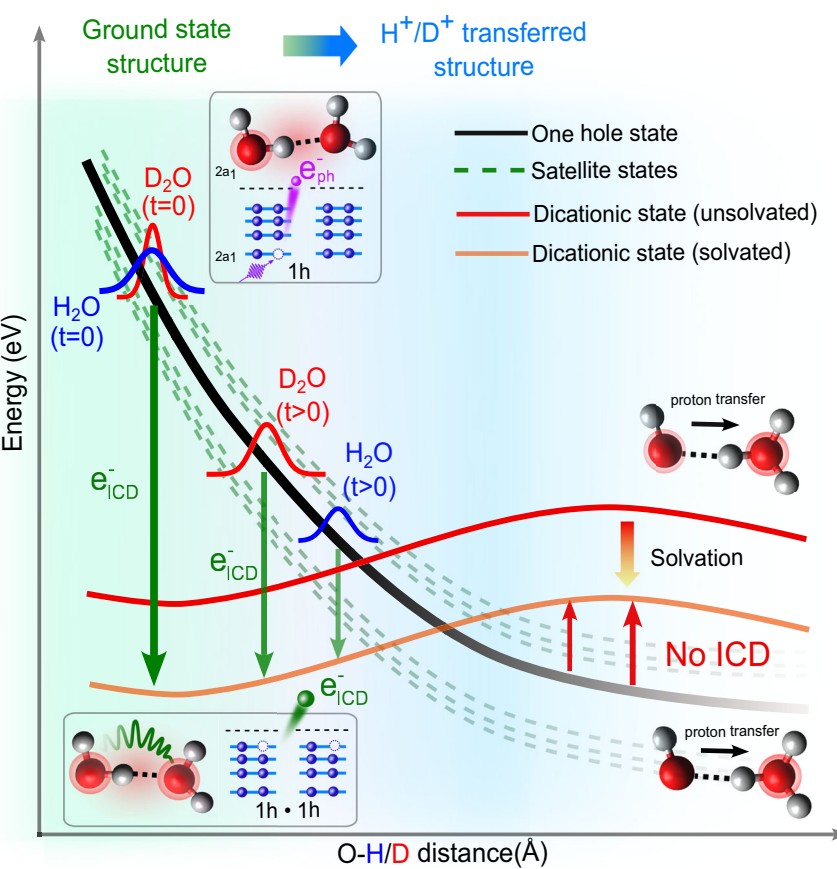

**Fig. 3 | Schematic representation of intermolecular coulombic decay and proton transfer in liquid water.** The black line represents the $(2a_1)^{-1}$ state and the dashed green lines the corresponding satellite states. The red and orange lines represent the unsolvated and solvated 1h-1h states, respectively.

the significant mixing of the main and satellite ionization states. Inner-valence ionization thus leads to the population of a band of states of comparable spectroscopic factors rather than that of a single state. For the water dimer, this effect is demonstrated in Fig. 4, which shows the states associated with the $2a_1$ ionization along the proton-transfer coordinate, calculated using the ADC(2,2) method[42]. This method is particularly suitable for scenarios involving a breakdown of the MO picture due to its ability to consistently describe the main and satellite ionization states.

Around the equilibrium geometry of the dimer, $2a_1$ ionization populates a band of states spanning a width of ~2 eV and initiates the proton-transfer dynamics. With increasing O-H distance, the band spreads up to about 5 eV beyond 1.5 Å. Around $R_{O-H} = 1.3$ Å, the lower part of the band drops below the lowest double-ionization potential (cf. also Table 1), and the ICD channel partially closes. However, in contrast to the picture presented in ref. 24 (cf. Fig. 3 therein), part of the inner-valence ionization band stays above the ICD threshold for all configurations. Nevertheless, even the more complete description

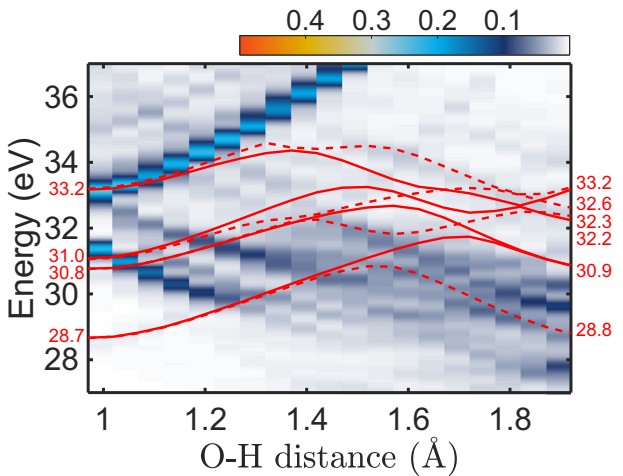

**Fig. 4 | Calculated potential-energy curves (PECs) of singly- and doubly-ionized water dimer along the proton transfer coordinate.** The intensity map shows pole strengths of the singly-ionized states in the inner-valence spectral region, manifesting a breakdown of the molecular-orbital picture. Solid and dashed red lines correspond to the few lowest-lying singlet and triplet doubly outer-valence-ionized states, respectively. The energies of the PECs are indicated in red. The configuration of the PECs are labeled in Table 1.

confirms the proton transfer as the cause of the reduced ICD efficiency.

The dimer is the smallest entity in which ICD occurs. It is, however, a special case as we deal with asymmetric proton donor and proton acceptor units. The immediate question is how the mechanism would be modified through further solvation. The simulations are complicated by two factors: (i) for a droplet of size $N$, we deal with $N$ $2a_1$ states and $N \cdot (N-1)/2$ doubly ionized states with a HOMO hole and (ii) one should take into account structural fluctuations in the liquid phase. Direct dynamical simulations would therefore be very demanding as we cope with non-adiabatic dynamics for a very large and dense manifold of electronic states.

To illuminate the effects of solvation on the ICD efficiency, we have applied the efficient QM:QM fragmentation technique. We started with a very large water droplet (5000 water units), simulated at 300 K with path-integral molecular-dynamics[43,44] simulations accounting for zero-point effects. We then selected a central water dimer (reaction zone) that was surrounded by an increasing number of solvating water units. The electronic and nuclear processes affecting ICD were further assumed to be limited to the reaction zone. Based on the dynamical simulations for isolated water dimer, we have identified two main dynamical coordinates: the first coordinate was the elongation of the O-H bond from a hydrogen-bond (HB) donor, the second one was the elongation of the second O-H bond of the HB donor. The first elongation was typically about twice as large as the second one. We have then performed energy scans for these coordinates, keeping all of the remaining atoms fixed.

Figure 5 shows the calculated energies along the first reaction coordinate (RC1) for 20 representative structures. The averages thus represent a potential of mean force for the $2a_1$ and doubly ionized states. In the equilibrium geometries, the $2a_1$ energies are safely above the energies of the doubly ionized states even for the water dimer, with an energy difference of about 2 eV. The decrease in the ionization energy upon the gradual solvation is much larger than for the $2a_1$ state. For the largest droplet considered (216 water molecules, see Fig. 5(f)), the energy difference amounts to ~6 eV. These values span the kinetic-energy range of the ICD electrons typically observed in the experiment (see also Fig. 1(a)).

However, both the energies for the $2a_1$ and doubly-charged states change with the proton transfer/relaxation RC1 coordinate and they do so in opposite directions. The singly-ionized state loses its energy as the system approaches the $H_3O^+$...OH adiabatic state, the doubly ionized state exhibits a high barrier at about 1.5 Å as a result of the interaction between the $H_2O^+$...$H_2O^+$ and $H_3O^+$...$OH^+$ diabatic states. For

## Table 1 | Character of the dicationic states shown as red lines in Fig. 4

| Geometry | Energy (eV) | Singlet | Triplet |
|---|---|---|---|
| Equilibrium (A: acceptor; D: donor) | 33.2 | $(HOMO)^D(HOMO)^A$ | |
| | 31.0 | $(HOMO-1)^D(HOMO)^A$ | |
| | 30.8 | $(HOMO)^D(HOMO-1)^A$ | |
| | 28.7 | $(HOMO-1)^D(HOMO-1)^A$ | |
| Proton Transfer (A: $H_3O^+$; D: $OH^+$) | 33.2 | $(HOMO)^A(HOMO)^D$ | |
| | 32.2 | $(HOMO-1)^D(HOMO-1)^D$ | |
| | 30.9 | $(HOMO-1)^D(HOMO)^D$ | |
| | 30.9 | $(HOMO)^D(HOMO)^D$ | |
| | 33.2 | | $(HOMO)^A(HOMO)^D$ |
| | 32.6 | | $(HOMO-1)^D(HOMO-2)^D$ |
| | 32.3 | | $(HOMO)^D(HOMO-2)^D$ |
| | 28.8 | | $(HOMO)^D(HOMO-1)^D$ |

For the equilibrium geometry, both the singlet and triplet states correspond to $H_2O^+ - H_2O^+$ and have the same energy within the quoted accuracy. For the proton transfer geometry, the lowest three singlet and triplet states are $OH^+ - OH_3^+$, the highest, fourth state is $OH - OH_3^{2+}$.

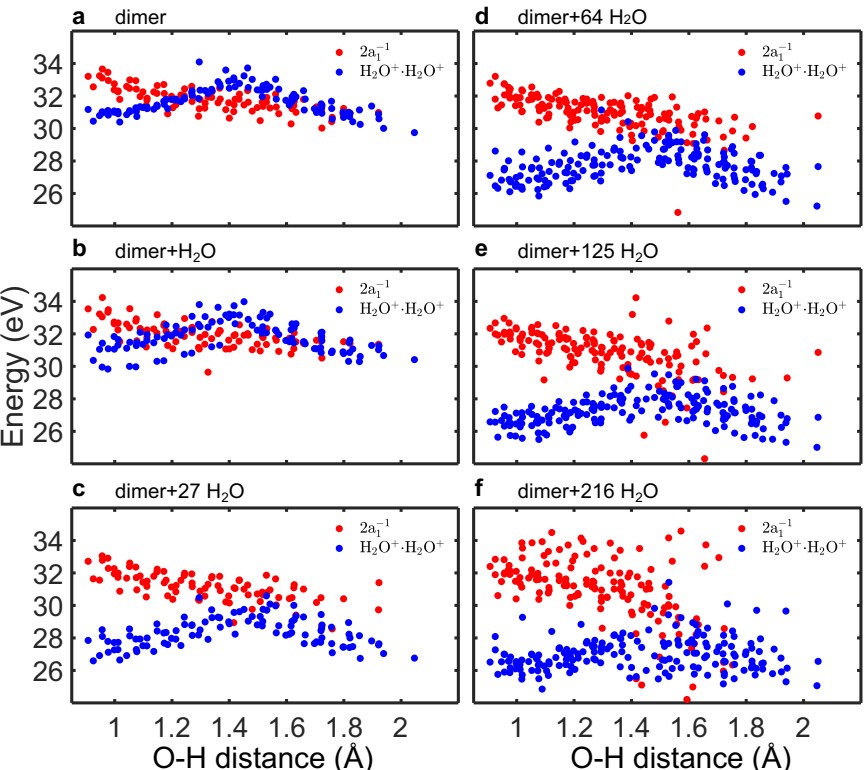

**Fig. 5 | Calculated energies of the singly ionized $2a_1$ (red) and doubly ionized cationic states (blue). a** Calculated energies of the singly ionized $2a_1$ (red) and doubly ionized cationic states (blue) for water dimer. **b–f** Calculated energies for water dimer associated with 1, 27, 64, 125 and 216 water molecules, respectively.

The solvent effects are modeled by increasing the number of the associated water molecules from 1 to 216. The oscillating structure in the data are an artifact resulting from the variations in the initial OH bond distances.

an isolated dimer, the two states start to cross almost immediately, at the O-H distances of about 1.2 Å, and the ICD channel then closes. The crossing distance increases with an increasing number of solvating water units, yet even for the largest cluster, we observe many instances when the ICD channel is closed. The scenarios described above remain valid also for the second reaction coordinate (RC2), with two hydrogen atoms moving simultaneously. While the energy changes are more pronounced both in the singly and doubly ionized states, the process remains qualitatively the same. These data are shown in the Supplementary Information.

The above results indicate that the ICD-decay channel is gradually closing upon the spontaneous proton-transfer dynamics. Therefore, solvation should increase the ICD efficiencies and the ICD efficiencies should be related to the kinetic-energy distribution. Unfortunately, direct ab-initio simulations of the ICD process in liquid water are not technically feasible. We have, therefore, developed a multi-scale stochastic model, combining the ab-initio molecular-dynamics simulations for a dimer with the energetics for the solvated system and empirical inclusion of the ICD and non-adiabatic rates. The aim is to accurately replicate the energy distribution of ICD electrons as observed in[24] and in the present work, shown in Fig. 6(b), the ICD efficiencies for gradually solvated system and the isotope effects.

We use the energy data from Fig. 5 to model the experimentally observed kinetic energy distributions of the ICD electrons. First, we focused on modeling the smallest cluster, consisting of a single dimer. In this case, the dynamics of the system on the $2a_1$ potential energy surface can be calculated explicitly. The system evolves on the $2a_1$ potential and it decays with an intrinsic lifetime $\tau$ to one of the final dicationic states. In our model, the doubly-ionized state was assumed to occupy not only the ground state but also one of five possible singlet and five triplet states. The non-adiabatic transitions are not considered

in this model as the ICD channel closes very soon and only rarely reopens (in less than 1 %). The results are presented in Fig. 6(a). The kinetic energy distribution curves are consistent with the experimental data on the smallest clusters (see Fig. 6(b)). The sensitivity to the ICD lifetime $\tau$ is negligible. Note that water dimer is exceptional because it features a unit that only acts as hydrogen-bond acceptor. Its inability to transfer one of its protons should lead to an increase in ICD efficiency. However, starting from the trimer, the PT deactivation mechanism is always open for all units of stable larger clusters because they all act as hydrogen donors.

Simulating larger clusters prompted us to employ a more advanced Monte-Carlo simulation. We now consider a dimer in the reaction zone embedded by the solvating molecules. The distribution of the OH distances is assumed to be the same as in the isolated dimer while the energetics of the singly and doubly ionized states in the solvated environment is taken from the data presented in Fig. 5. Both the geometries at each state and the energies are sampled from the corresponding distributions. This method necessitated an additional parameter, $\delta$, which accounts for the systematic, time-dependent decrease in the energy of the singly-ionized state. The physics behind this parameter is the additional energy reduction due to the non-adiabatic decays of the singly-ionized state in a dense manifold of electronic states. The outcomes of this simulation are displayed in Fig. 6(a). The kinetic energy distributions reproduce the distributions measured experimentally in Fig. 6(b).

The Monte-Carlo simulation also provides ICD efficiency data, shown in Fig. 6(c), where specific combinations of $\delta$ and $\tau$ are found to match the experimental data[24] (the fitting procedure is described in more detail in the SI). Notably, the best fit is achieved with $\tau$ set at 50 fs and $\delta$ to 0.3 eV/fs. The relatively small value of $\delta$ appears to be physically reasonable, as it corresponds to modest energy decreases.

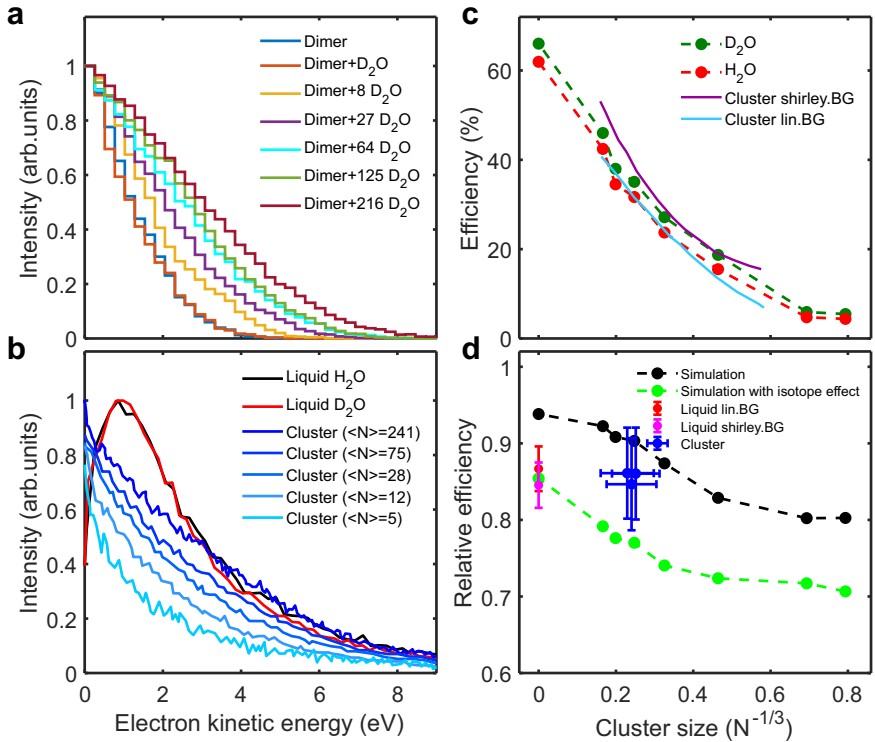

**Fig. 6 | The energy distributions of ICD electrons and the efficiencies of ICD in H₂O and D₂O (clusters and bulk). a** Energy distributions of ICD electrons with increasing number of D$_2$O units. The simulations were done using a Monte-Carlo approach, considering proton transfer in the inner-valence ionized state followed by ICD decay (time constant $\tau$) and additional energy drop in the inner-valence-ionized state ($\delta$). The intensity is normalized to its maximum. **b** Comparison of ICD-electron spectrum with liquid and cluster results. **c** Simulated ICD efficiencies for H$_2$O (red) and D$_2$O (green) at different cluster sizes. The experimental results of water clusters were extracted from ref. [24] for comparison. The ICD lifetime ($\tau$) and energy relaxation rate of the singly-ionized state ($\delta$) were chosen as $\tau$ = 50 fs, $\delta$ = 0.3 eV/fs, respectively. **d** The relative ICD efficiencies of H$_2$O compared to D$_2$O with (black dot) and without (green dot) isotope effect. The liquid results obtained by two different background-subtraction methods are shown for comparison. The blue dots represent the relative ICD efficiencies of water clusters[24]. Error bars shown here represent the standard deviation due to stochastic errors.

Furthermore, this method qualitatively captures the relative ICD efficiency of H$_2$O compared to D$_2$O, as depicted in Fig. 6(d).

The majority of the emitted electrons originate from the ICD process. However, a detailed analysis reveals that a fraction of the electrons arises from highly stretched configurations, corresponding to OH$^\bullet$...H$_3$O$^+$ structures. The autoionization of these states can be classified as a local Auger decay. This overall mechanism closely resembles the proton-transfer-mediated decay previously described for hydrogen-bonded systems following core-electron ionization[45,46]. The contribution of these decay events is 4.5% for the largest cluster considered, and for regular water, the contribution is smaller for smaller clusters and deuterated water. These estimates are based on using the same effective lifetime $\tau$ for both ICD and Auger-decay-like transitions. This assumption is consistent with Fano-ADC(2,2)[42] calculations for the dimer, which predict effective lifetimes of the 2a$_1$ band of 41 fs and 52 fs for the equilibrium and proton-transfer configurations, respectively.

## Discussion

The phenomenon of non-local decay processes, like Intermolecular Coulombic Decay (ICD), has garnered interest in recent years, particularly for its implications in radiation chemistry. These processes are significant because they release low-energy electrons that can potentially damage biological systems. However, their role in radiation chemistry is still poorly understood, largely due to limited data on the probabilities of these processes and their subsequent dynamics.

Water, as the most critical medium in radiation chemistry, serves as an ideal starting point for deeper investigation. Our study specifically focuses on the ICD process following the ionization of the 2a$_1$ state in water clusters and liquid water. The findings demonstrate that

although ICD is an important pathway, its efficiency remains measurably below unity—this remained an open question in earlier studies that indicated an increase in efficiency with larger cluster sizes[24].

Measured kinetic-energy distributions reveal that the 2a$_1$ state initially lies well above the lowest doubly ionized state, with an energy gap between 6 and 8 eV. We modeled the solvent effects across cluster sizes, showing that the gap decreases along the proton-transfer coordinate. However, if proton transfer were the only mechanism at play, ICD efficiency should approach unity, which is not observed.

This discrepancy led us to hypothesize additional non-adiabatic energy relaxation mechanisms. Our study emphasizes the complexity of the 2a$_1$ one-hole state, which is embedded in multiple satellite states that can interact via non-adiabatic transitions.

The present study also suggests that a smaller fraction of the electronic decays can be classified as local proton-transfer-mediated decay. The final product would then be OH$^+$ and H$_3$O$^+$.

The data we gathered align with an ICD lifetime of ~50 fs and a non-adiabatic relaxation rate of 0.3 eV/fs, consistent with prior calculations on similar systems like neon[1,47] and water complexes[24]. While the current model does require fitting two parameters to match experimental data, it remains robust across a wide range of cluster sizes. Consequently, it provides a reliable foundation for incorporating ICD into radiation chemistry models, particularly in biological environments where liquid water plays a crucial role.

## Methods

### Experimental methods

The experiments were performed by combining a monochromatized table-top high-harmonic-generation (HHG) source, a liquid micro-jet

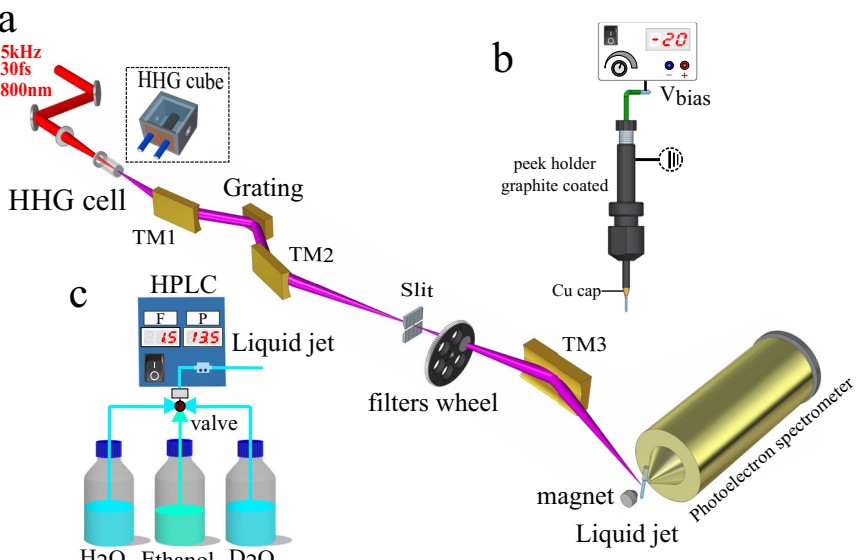

**Fig. 7 | Schematic illustration of the experimental setup. a** Scheme of the XUV-monochromator, adapted from Fig. 1a in ref. 31. **b** Illustration of how the bias potential is applied to the liquid jet. **c** Illustration of the liquid system allowing for switching between liquid $H_2O$ and $D_2O$ under identical experimental conditions.

and an electron-electron coincidence photoelectron spectrometer, as shown in Fig. 7(a). The extreme-ultraviolet (XUV) pulses were delivered by a time-preserving monochromator. High-harmonic generation was driven by a near-infrared laser pulse of ~1.2 mJ and ~30 fs duration centered at 800 nm with a repetition rate of 5 kHz. For the XUV energies below 60 eV, the driving pulse was focused into a semi-infinite gas cell filled with 25 mbar of neon, and the generated harmonics were collimated by a toroidal mirror (TM1) and spatially diffracted by a 300 lines/mm grating mounted in conical-diffraction geometry. For the generation of photon energies above 60 eV, instead of the semi-infinite gas cell, we used a 3 mm diameter metallic tube filled with 19 mbar of neon to extend the cut-off energy of the HHG spectrum. Accordingly, a higher groove density grating (600 lines/mm) was introduced to efficiently separate the generated harmonics. The diffracted harmonics were then focused by a second toroidal mirror (TM2) onto a 50 μm slit to select a single harmonic order. Finally, the XUV beam containing only a single harmonic order was imaged by a third toroidal mirror (TM3) onto the liquid micro-jet in the interaction chamber.

Liquid water was delivered into the chamber through a 25 μm inner-diameter quartz nozzle, which was capped with Cu tape, held together by Sn solder to prevent the insulating quartz from charging up due to stray electrons. The nozzle mounting system is made of PEEK and graphite-coated to ensure electrical conductivity (see Fig. 7(b)). The electrokinetic charging effect of the jet was minimized by adding NaCl at a concentration of 50 mmol/L. Moreover, a bias voltage of 0.45 V was applied to the liquid jet to simultaneously compensate the effects of the residual streaming potential and that of the vacuum-level offset between the jet and the photoelectron spectrometer[35]. A commercial HPLC pump from WATREX Prague (Model: P102) was utilized to transport the liquid sample into the chamber, the flow rate was set to 1.5 mL/min, while the stagnation pressure was maintained as 13.5 bar. In order to maintain identical experimental conditions, for each measurement at the same XUV energy, the liquid $H_2O$ and $D_2O$ were switched via a three-way valve immediately after one measurement was finished (see Fig. 7(c)).

The electrons emitted from the liquid jet were recorded by a magnetic-bottle photoelectron spectrometer, previously described in[48], consisting of a permanent magnet (1 T), holding a conical iron tip, a 910 mm-long flight tube equipped with a solenoid that generates a

homogeneous magnetic field of 1 mT along the flight tube. In this configuration, electrons with a pitch angle between 0° and ~120° are collected, corresponding to a solid angle of $2.8\pi$ sr. Without applying any bias potential on the skimmer and the flight tube, the spectrometer was first calibrated by photoionization of argon with various harmonics (H11–H51), the XUV energy was calibrated to an accuracy of ±0.06 eV. The average energy resolution of the spectrometer was estimated as ≈55:1 for the measured electrons in this study.

It is well known that the detection efficiency of a magnetic bottle spectrometer varies for photoelectron kinetic energy below ~1 eV[49] and remains relatively constant for photoelectron kinetic energies between 1 and 100 eV[50]. Therefore, in order to enhance the detection efficiency of the slow electrons (<1 eV), a bias potential (+0.9 V) was applied both on the skimmer and the flight tube. The measured spectra were corrected for the applied potential. The raw MCP signal was amplified 10 times by a home-built fast pre-amplifier and then recorded by an analog-to-digital (ADC) converter. To sufficiently suppress the false-coincidence events, the waveforms were collected under extremely low count rates (~0.08–0.09 counts/laser pulse). The double-hit capability was determined by the 0.5-ns time resolution of the ADC device. For each coincidence map, the spectra were recorded over 40 million laser shots with typical data-acquisition times of 135 min. Applying an optimized threshold to reduce the contribution from electronic ringing, the measured raw waveforms were then processed and sorted into single, double and multi-hit (≥3) events.

## Theoretical methods

**The QM:QM fragmentation approach.** Calculating the energies of molecules within large clusters presents significant challenges. Traditional quantum chemistry methods are often impractical due to their computational requirements or conceptual problems with highly excited states.

To address these challenges, we employed a stable and accurate fragmentation method that approximates the system by treating each molecule as an individual fragment[51]. For each fragment, we calculated the static point charges of its atoms. Once the charges were determined, a fragment was selected and surrounded by the point charges of the other fragments, representing the solvation effect. The energy of this fragment was then computed. This process was repeated for every fragment in the system, after which the total energy was

obtained by summing the individual fragment energies. Note that the energy calculation for each fragment is influenced by the surrounding charges. Consequently, we can refine the point charges iteratively, leading to a self-consistent cycle that continues until the energy reaches a satisfactory level of precision. The final expression of the total energy, $E_{total}$, is given as:

$$E_{total} = \sum_{i=1}^{n} E_i - \sum_{A} \sum_{A<B} \frac{Q_A Q_B}{R_{AB}} \quad (2)$$

where $i$ denotes the fragment index, $E_i$ is the energy of the $i$th fragment, $n$ is the total number of fragments, $Q_A$ and $Q_B$ are the charges of atoms A and B, and $R_{AB}$ is the distance between these two atoms. The second term in equation (2) subtracts the potential energies between atoms belonging to different fragments, thereby avoiding the double counting of these interactions.

This method defines two regions: one representing the solvent and the other representing the central molecule or cluster. Both regions can be treated using the same or different levels of theory. The fragmentation approach significantly reduces computational time while maintaining high accuracy, even for very large systems, without sacrificing the granularity of the solvent.

In this study, we focused on a pure water cluster, aiming at calculations of energies for two distinct states of the water dimer in the central reaction zone. The first state corresponds to the energy of the $2a_1$ inner-valence ionized state of the dimer, where the electron hole was generated using the Maximum Overlap Method (MOM)[52]. The second state corresponds to the energy of the doubly ionized dimer.

To construct the potential-energy curves, we elongated the OH bond of the initially ionized water molecule, which participates in proton transfer. This bond was stretched from -1 Å to 2 Å, while the second OH bond of the same water molecule was elongated by half the distance of the first bond. During this process, only the hydrogen atoms were allowed to move, with the remaining atoms in the dimer held fixed. By plotting the total energy of the ionized dimer as a function of the OH bond length and subtracting the total energy of the ground state (with no bond elongation), we obtained the potential energy curves. The kinetic energy was then calculated as the difference between the potential energy curves of the doubly ionized state and the $2a_1$ ionized state. We also modeled the influence of cluster size on the behavior of these potential curves, treating the dimer as the core region and the solvent as a collection of additional water molecules.

The calculations were performed using Q-Chem 5.4 with the MP2 method, employing the aug-cc-pCVTZ basis set for oxygen and aug-cc-pVTZ for hydrogen[53]. We generated twenty different potential energy curves for various cluster sizes. However, convergence issues led to missing data points, complicating the computation of kinetic-energy values, which require matching points from both the $2a_1$ ionization and double-ionization states at the same OH bond length.

Initial geometries were sampled using quantum molecular dynamics simulations performed with our in-house software, ABIN version 1.1[54]. The simulation temperature was set to 300 K and maintained using a Generalized Langevin Equation (GLE)[44] thermostat. The time step was set to 40 atomic units, with a maximum of 100,000 steps. The first 10 ps of the simulation were treated as the thermalization phase, and twenty geometries were selected evenly from the remaining simulation time.

**Modeling kinetic energy distributions and ICD efficiencies.** To model the distribution of ICD-electron kinetic energies in large clusters, several key assumptions must be established. Our approach is to utilize geometries of the water dimer, as obtained from molecular simulations in a vacuum. In this context, we assume that the motion of the central dimer is largely independent of the cluster size. The geometry of the dimer evolves over time, described by its trajectory, which

is characterized by two primary variables: time and the hydrogen-oxygen distance.

Initial conditions for these trajectories were generated using molecular dynamics simulations, performed with the ABIN[54] code, while electronic energies were computed using the TeraChem software. The dynamical calculations were executed on the PBE/6-31++G** potential energy surface. The system's temperature was controlled at 250 K using a GLE thermostat[44]. The time step was set to 20 atomic units (au), and the simulations were run for a maximum of 100,000 steps. The first 10 ps of the simulation were dedicated to thermalization, and from the remaining time, one hundred initial conditions were evenly sampled for further analysis.

The dynamics in the inner-valence ionized state was generated through classical molecular-dynamics simulations using the ABIN[54] code, employing the MOM as implemented in the Q-Chem version 6.0[53]. These simulations were performed in a microcanonical ensemble. The time step was set to 10 atomic units (au), with a maximum of 500 steps, though some trajectories were cut short due to convergence issues. Despite these challenges, we successfully obtained one hundred distinct trajectories for analysis.

To estimate the ICD-electron kinetic energy based on the dimer trajectories, we utilized potential energy curves derived from fragment-based methods. The kinetic energy was determined by the difference between the potential energy curves of the $2a_1$ state and the doubly ionized state. We generated twenty such potential energy curves, which were then fitted to a fourth-degree polynomial. Additionally, we fitted the standard deviation of the kinetic energy points using also a fourth-degree polynomial to account for variations.

In summary, while each trajectory exhibits unique characteristics, the overall kinetic energy function remains consistent for a given cluster size. Before implementing the Monte-Carlo algorithm, it is necessary to describe the energetic processes and corrections that influence the overall system behavior.

In the first assumption, we consider that the doubly-ionized state can decay into one of eleven possible states. The first state is the triplet ground state, while the next five are triplet excited states with energy shifts of 2.25, 2.3, 4.6, 6.0, and 8.0 eV, respectively, from the lowest to the highest energy levels. For simplicity, we assume these energy shifts remain constant, regardless of changes in the O-H bond distance. The remaining five states belong to the singlet manifold, with energies -0.5 eV higher than their corresponding triplet states.

The second assumption introduces a time-dependent function to model the energy reduction of the singly-ionized state due to non-adiabatic interactions. This energy decrease is represented by a simple linear function, $E_{drop} = \delta \cdot t$, where $t$ is time and $\delta$ is a constant that defines the rate of energy loss over time.

We can now construct the overall energy balance and corrections as described in equation (3). The kinetic energy, $E_{kinetic}$, is sampled from a normal distribution, where the mean value is determined by the fitted kinetic energy function, and the standard deviation is derived from the second fitted function. The term $E_{state}$ accounts for the energy correction corresponding to the singlet or triplet states of the doubly-ionized species, while $E_{drop}$ represents the systematic decrease in energy over time, governed by the constant $\delta$.

To estimate the energy in the bulk phase, $E_{bulk}$ is set to 2 eV, and we use the kinetic energies corresponding to the largest available cluster size of 216. If the bulk phase is not considered, $E_{bulk}$ is set to zero.

$$E_{kinetic} = E_{distribution} - E_{state} - E_{drop} + E_{bulk} \quad (3)$$

Next, we describe the algorithm. We start at time zero when the $2a_1$ electron-hole forms. From this point, we proceed in 0.24 fs time increments. At each step, we check whether the kinetic energy, $E_{kinetic}$, is positive. If the kinetic energy is positive, we randomly select one final

state (if multiple states with positive $E_{kinetic}$ exist). With a probability equal to $\frac{\Delta t}{\tau}$, where $\tau$ is the parameter representing the ICD lifetime and $\Delta t$ is the time step of 0.24 fs, ICD-electron emission occurs, and we record the kinetic energy. If no electron emission occurs, the trajectory continues to the next step. If the kinetic energy becomes negative, the trajectory may still proceed to a region where the ICD channel reopens, restoring positive kinetic energy and enabling electronic decay.

For better statistical accuracy, we run each trajectory one thousand times, resulting in a total of one hundred thousand potential ICD-electron emissions. This allows us to calculate the efficiency as the ratio between the number of emitted electrons and the maximum number of possible trajectories.

The dimer is the only system for which we can fully model the distribution of ICD electrons because it allows us to directly compute the kinetic energy at every point along each trajectory. For this purpose, we employed the QM approach with the same settings used for the potential energy curves. The algorithm is essentially the same as the one used in the Monte Carlo simulations above, but in this case, the kinetic energies are directly obtained from the QM calculations at each point along the trajectory. In this simulation, we did not account for the energy decrease of the singly-ionized state as all the trajectories become inactive very fast.

### Reporting summary
Further information on research design is available in the Nature Portfolio Reporting Summary linked to this article.

## Data availability
All the data that support the findings of this study are available in the main text and the Supplementary Information, or from the corresponding authors P.Z., H.J.W., or P.S. upon request. Source data are provided with this paper. Results from electronic-structure and dynamics simulations are given as Supplementary Data 1. Source data are provided with this paper.

## Code availability
The code that supports the findings of this study is available from the corresponding authors upon request.

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

## Acknowledgements

This work was supported by ETH Zürich. P.Z. acknowledges the support by the National Natural Science Foundation of China No. 12474261. J.D. and P.S. were supported by the Czech Science Foundation (EXPRO project no. 21-26601X). This work was supported by the project "The Energy Conversion and Storage", funded as project no. CZ.02.01.01/00/22 008/0004617 by Programme Johannes Amos Comenius, call Excellent Research. P.K. acknowledges financial support by the Czech Science Foundation (Project GAČR No. 22-22658S).

## Author contributions

H.J.W. and P.Z. conceived the study. P.Z. performed the experiments with the support of J.T. P.Z. analyzed the data. J.D., P.K., and P.S. performed the calculations. All authors discussed the data and contributed to writing the manuscript. H.J.W. supervised the realization of the experiments. P.S. supervised the simulations.

## Competing interests

The authors declare no competing interests.
