## [Transparent Peer Review file · Nature Communications]

Intermolecular Coulombic decay in liquid water competes with proton transfer and non-adiabatic relaxation

Corresponding Author: Professor Hans Jakob Wörner

Version 0:

Reviewer comments:

Reviewer #1

(Remarks to the Author)

The report is attached as a PDF.

Reviewer #2

(Remarks to the Author)

In their manuscript, Zhang et al. report an experimental–theoretical study of intermolecular Coulombic decay (ICD) in liquid water, investigating its efficiency. ICD is a non-local decay mechanism that occurs in weakly bound media (typically after inner-valence ionization) and simultaneously produces several genotoxic species (such as low-energy electrons and radical cations) in the immediate vicinity of the primary ionization site. Thus, it is highly relevant to the radiation damage of bio-matter and has been intensively studied since its theoretical prediction in the late 1990s. The process is typically extremely efficient, with lifetimes in the range of a few tens of femtoseconds. ICD in water has been the subject of a number of theoretical and experimental studies. Earlier works on water dimers and small water clusters have shown that, when initiated by the inner-valence ionization of the proton-donating molecule, ICD competes with the proton-transfer process, which can energetically close the decay channel. However, when comparing the ICD efficiency of water clusters of different sizes, it has been suggested (Ref. [24]) that the efficiency of the process in liquid water should approach 1, due to a combination of effects that prevent channel closing along the proton-transfer coordinate. By performing measurements in liquid regular (H₂O) and heavy (D₂O) water, as well as multi-scale stochastic model calculations, the present work tests this assumption, showing that even in bulk water the ICD efficiency remains well below 1 -- practically as in a cluster with a few tens of water molecules. Although an absolute efficiency cannot be extracted, a careful analysis of the experimental data shows a relative efficiency (H₂O/D₂O) of 86%, which is in line with the value determined for water clusters with a mean size of 72 molecules (Ref. [24]). The reported simulations suggest that channel closing by proton transfer cannot explain the observed efficiency; instead, other non-adiabatic processes occurring within the band of states populated by the ionization of a 2a₁ electron should also contribute. Although the present work does not provide concrete ideas for such mechanisms, I find the results very interesting and with the potential to spark further studies on this fascinating topic.

The manuscript is clearly written, and the analysis is carefully executed. Understanding non-local decay mechanisms in aqueous systems is important not only for radiation damage but also in a broader context, making the reported results highly relevant to a wide audience. Therefore, I support the publication of this work in Nature Communications. However, there are several issues that the authors should address before I can recommend publication.

1) In my opinion, the main finding of this study is that proton transfer is not the sole mechanism responsible for the observed reduced ICD efficiency in liquid water. This point, however, is not emphasized enough in the manuscript. It is actually mentioned only briefly in the Discussion and Conclusions section. The authors state, "This discrepancy led us to hypothesize additional non-adiabatic energy relaxation mechanisms", yet no such mechanisms are discussed in the manuscript. The authors should elaborate on this point and propose a plausible mechanism. Moreover, in the Abstract, the proton transfer is still presented as the only explanation.

2) In relation to the previous point. I find the ADC(2,2) results reported in Fig. 4 very important, as they demonstrate that one cannot consider a single 2a₁ state and potential, but rather a band of non-adiabatically coupled states spread over several eVs. Isn't this the key to understanding the complex non-adiabatic dynamics initiated by inner-valence ionization? It is not

clear from the description of the computational scheme how the existence of the 2a1 band has been incorporated into the semi-classical simulations. The authors continue to talk about a single 2a1 potential (see, e.g., page 10).

3) When deriving the equation for ICD efficiency, the authors state on page 4 of the SI that the ICD process can occur only in the liquid phase, implicitly assuming that no dimers (or larger clusters) form in the vapor surrounding the jet. This assumption is not self-evident to me. Can the authors provide further arguments to support it?

4) In Fig. 6 (panels b and d), the authors report the ICD efficiency for the dimer as being well below 10%. Although this might be the case if ICD is initiated by ionizing the proton-donor molecule, the ionization of the proton acceptor should lead to decay with very high efficiency since no proton transfer is initiated and the decay channel remains open. Given that the probability of ionizing either molecule in the dimer is 50%, one would expect the overall efficiency in the dimer to exceed 50%. This situation, of course, changes in larger clusters where every molecule simultaneously acts as both proton donor and proton acceptor. The authors should either explain these results or remove the dimer from the figures reporting efficiency trends.

A few more minor issues:

1) In Fig. 2 (and 6) the authors report results from two different background subtraction methods. Although details are certainly not needed, what is meant by "lin.BG" and "shirley.BG" should somehow be pointed out in the caption.

2) On page 7 of the manuscript, the authors discuss the breakdown of the molecular-orbital picture, giving rise to the population of a multitude of states by removing an electron from the 2a1 orbital. The authors should cite some of the original works on the subject.

3) The authors should include the color bars in Fig S1.

4) First sentence of the last paragraph on page 16: "... the doubly-ionized states can decay into...". The authors likely mean that the singly-ionized states can decay to eleven doubly-ionized states.

5) The redundant letter "a" should be removed from panel e of Fig. S2.

Reviewer #3

(Remarks to the Author)

The manuscript "ICD in liquid water competes with proton transfer and non-adiabatic relaxation", the authors report a coincidence photoelectron spectroscopy study of electronic and nuclear dynamics following ionization of liquid water with 63.5 eV photons. The experiments extend the work recently published in Ref. 31 by the experimental team to compare the dynamics of the heavy hydrogen atoms in D2O with the previously-reported H2O results. The data are interpreted with the aid of quantum chemistry electronic structure calculations, ab initio molecular dynamics simulations of water clusters, adapted to the electronic structure of liquid water with empirical ICD and nonadiabatic relaxation rates. The authors also pointed out the experimental challenges and the importance of studying ICD to understand many aspects of secondary or low-energy electrons relevant to modeling radiation damage. They claim from their experimental results and theoretical simulations that proton transfer and non-adiabatic relaxation are the main reasons for ICD, which contrasts with the ref [24]. Overall, the results reported in the manuscript are important incremental advances on recently published closely related work, specifically Ref. 31 and Slavicek et al. [dx.doi.org/10.1021/ja5117588](https://doi.org/10.1021/ja5117588), J. Am. Chem. Soc. 2014, 136, 18170–18176).

The presence of low energy electrons measured in coincidence with inner-valence (2a1) photoelectrons is clearly consistent with ICD. However, local autoionization processes, such as Auger-Meitner decay may also compete with ICD. Further analysis of local autoionization processes is needed to justify the assignment of the spectral region between the red lines in Fig. 1 b and d as "... dominated by 2a1-photoelectron/ICD-electron pairs". To accurately determine the ICD efficiency, ICD must be isolated from all other ionization processes. How is local autoionization excluded? The authors must address this question before the manuscript will be suitable for publication.

In addition the above primary concern, the color scale should be added to Figure S1.

Reviewer #4

(Remarks to the Author)

Reviewer #5

(Remarks to the Author)

see the attached file

Version 1:

Reviewer comments:

Reviewer #1

(Remarks to the Author)

The revised manuscript addresses all my suggestions. I am glad to recommend publication of this manuscript in nature communications

Reviewer #2

(Remarks to the Author)

The authors have thoroughly addressed all of my previous criticisms and recommendations, as well as those raised by the other reviewers. They have revised the manuscript by extending the discussion and clarifying several key aspects of their approach and analysis. In addition, they have performed new simulations and refined their model, which now shows even better agreement with the experimental data.

As I already stated in my initial report, I believe the manuscript presents results of general interest and contains important new findings that merit publication in Nature Communications. I therefore recommend the paper for publication.

I would like to point out one minor issue: in the revised abstract, the abbreviation ICD is defined twice, which is obviously unnecessary.

Reviewer #3

(Remarks to the Author)

In the revised manuscript, revised supplementary information, and responses to the referee reports, the authors have thoroughly addressed each of my comments and concerns, as well as those of the other referees. I recommend the manuscript publication in Nature Communications without further revision.

Reviewer #4

(Remarks to the Author)

Reviewer #5

(Remarks to the Author)

Overall, after not too many changes, the draft has somewhat improved.

I agree that the paper introduces a novel experimental technique to measure the relative efficiencies of Interatomic Coulombic Decay (ICD) in liquids, using electron-electron coincidence spectroscopy with liquid microjets. I also concur, that this work highlights the breakdown of the orbital approximation and the effects of solvation on potential-energy surfaces. It is commendable, that the study uses a multi-scale model that unifies results across liquid-phase and cluster systems. Yet, the reasons why the ICD efficiency in liquid water is not unity is still elusive. It appears Reviewer 2 and I share the same concerns in terms of delivering this breakthrough. Given the title, introduction, and overall aspiration of this paper, the hypothesized non-adiabatic energy relaxation mechanisms are (still) not well discussed (just mentioning the word "non-adiabatic" more frequently in the paper is not really the improvement I was after).

The authors now clarify that these mechanisms are represented in their model by "...an additional parameter, γ , which accounts for the systematic, time-dependent decrease in the energy of the singly-ionized state. The physics behind this parameter is the additional energy reduction due to the non-adiabatic decays of the singly ionized state in a dense manifold of electronic states." This is helpful, yet, insights into the non-adiabatic transitions from the satellite states that formed during the creation of the 2a1 vacancy and which are sketched in Fig. 3 to the calculated PECs in Fig. 4 are lacking. From the response to Reviewer 2, I now understand that this is not possible, because "the ADC(2,2) potentials are not explicitly used due to the unavailability of the necessary methodology, specifically the nonadiabatic couplings.". I think this will disappoint the general readership, as it leaves them with an answered problem.

If the draft goes forward to publication in Nat. Commun., which in my view it still does not really qualify for (see above), at a minimum the authors should clue in the general reader about this deficiency and dedicate a paragraph to explaining the current limitations, in order to put the speculations about non-adiabatic transitions in perspective and state the remaining open questions.

Some additional comments and suggestions:

- The new Fig. S2 and Table I are very helpful.

- I recommend adding a sentence or two on the answer about the decreasing detection efficiency towards zero kinetic energy in Section S.1, stating that this poses no problem for the measurement and the presented results, in order to put the reader at ease.

- Fig. 2 and S3 have improved. Yet, statistical error bars are still missing. If the authors don't want to explicitly show them, a comment on the uncertainty in the captions would also work. I also recommend to mention the approximate average energy resolution of the spectrometer (shown in Fig. R2) as ~55:1 in, e.g., the Experimental Methods section.

- Fig. 5 is still hard to read because of the scattered data points. If this cannot be smoothed out without compromising the calculated results, maybe at least mention that the oscillations blurring the data are an artifact resulting from the variations in the initial OH bond distances.

Response to the referees

We would like to thank all five referees for their valuable time they have taken to review our manuscript and their insightful comments and for their excellent suggestions, which helped us to further improve our manuscript. Below, we have color-coded each referee's comment in blue, our replies in black and additions/changes in green in our resubmission. We hope that the referees will find our answers and the revised manuscript satisfactory.

Reviewer #1 (Remarks to the Author):

This manuscript combines experiments, theoretical calculations and simulations to understand Intermolecular Coulombic Decay (ICD) in liquid water. The inner-valence $2a_1$ vacancy in liquid water, which is initially created using XUV photons, decays into dicationic state via ICD. With electron-electron co- incidence measurements they measured the kinetic energy of the photoelectrons ($2a_1$) and the ICD electrons in coincidence, thereby verifying the ICD in liquid water. Earlier Richter et al. (2018) have reported that the efficiency of ICD is below unity and the isotope effect observed by them pointed to the role of proton transfer in suppressing ICD. However, the calculations presented by Richter et al. (2018) did not indicate the suppression of ICD by proton transfer. This manuscript attempts to address this controversy through extensive and interesting experiments, calculations and simulations. They measure the efficiency of ICD in both regular and heavy water and observe a difference in the ICD efficiency for these two species. This enabled them to assert the role of proton transfer, as the dynamics of proton transfer is expected to proceed slowly in the heavy water, thereby rendering ICD more efficient in the deuterated water. Non adiabatic transitions as well strengthens ICD in deuterated water. Importantly, they study the effects of solvation. With increasing solvation they observe the closing of ICD channel to occur at larger reaction coordinate (along the proton transfer coordinate). Most importantly, despite considering the proton-transfer, the observed low efficiency of ICD was not accountable. The present manuscript stresses on the role of non-adiabatic transitions in critically determining the ICD efficiency. These non-adiabatic transitions occur due to the satellite states formed during the creation of the $2a_1$ vacancy. These results have implications on radiation damage initiated by low energy electrons in biological systems where water is the major medium. Hence, I am glad to recommend the manuscript for publication in *Nature Communications*.

We thank the reviewer for their accurate summary and for recommending acceptance of our manuscript.

I have a few questions and suggestions which are given below.

1. The ICD channel is found to compete with the proton transfer. If we consider water dimer (gas-phase) then the measurement of the kinetic energy of the two water ions formed from Coulomb explosion post ICD could shed light on the dynamics of proton transfer. Did the authors perform any measurements in this connection?

This is indeed an excellent idea. Experiments of this kind have been performed on water dimers to verify that ICD takes place (Ref. 22). In large clusters and in liquid water, such experiments are hampered by the impossibility to detect the two water ions because they remain trapped inside the cluster or liquid bulk, preventing their isolation and detection.

2. Figure 3, the cartoon at the lower left side of the figure includes both ICD and proton transfer process, which could confuse the readers. Only ICD cartoon could be retained. The proton transfer is anyway depicted along the reaction coordinate.

We thank the reviewer for this valuable suggestion, we have removed the arrow indicating the proton transfer and the phrase "proton transfer" at the lower left side of the cartoon.

3. Page 9: Please rewrite the sentence: ".....as a result of the interaction between the crossing between."

We thank the reviewer for recommending this correction.

We have removed the words “crossing between the”.

4. Page 10: Please rewrite the sentence: ”..... The aim is to accurately replicate....

We have rewritten this sentence to read:

“The aim is to accurately replicate the energy distribution of ICD electrons (as observed in [24] and in Fig. 6(b)), the ICD efficiencies for increasing degrees of solvation and the isotope effects.”.

Reviewer #2 (Remarks to the Author):

In their manuscript, Zhang et al. report an experimental–theoretical study of intermolecular Coulombic decay (ICD) in liquid water, investigating its efficiency. ICD is a non-local decay mechanism that occurs in weakly bound media (typically after inner-valence ionization) and simultaneously produces several genotoxic species (such as low-energy electrons and radical cations) in the immediate vicinity of the primary ionization site. Thus, it is highly relevant to the radiation damage of bio-matter and has been intensively studied since its theoretical prediction in the late 1990s. The process is typically extremely efficient, with lifetimes in the range of a few tens of femtoseconds. ICD in water has been the subject of a number of theoretical and experimental studies. Earlier works on water dimers and small water clusters have shown that, when initiated by the inner-valence ionization of the proton-donating molecule, ICD competes with the proton-transfer process, which can energetically close the decay channel. However, when comparing the ICD efficiency of water clusters of different sizes, it has been suggested (Ref. [24]) that the efficiency of the process in liquid water should approach 1, due to a combination of effects that prevent channel closing along the proton-transfer coordinate. By performing measurements in liquid regular (H₂O) and heavy (D₂O) water, as well as multi-scale stochastic model calculations, the present work tests this assumption, showing that even in bulk water the ICD efficiency remains well below 1 – practically as in a cluster with a few tens of water molecules. Although an absolute efficiency cannot be extracted, a careful analysis of the experimental data shows a relative efficiency (H₂O/D₂O) of 86%, which is in line with the value determined for water clusters with a mean size of 72 molecules (Ref. [24]). The reported simulations suggest that channel closing by proton transfer cannot explain the observed efficiency; instead, other non-adiabatic processes occurring within the band of states populated by the ionization of a 2a₁ electron should also contribute. Although the present work does not provide concrete ideas for such mechanisms, I find the results very interesting and with the potential to spark further studies on this fascinating topic.

The manuscript is clearly written, and the analysis is carefully executed. Understanding non-local decay mechanisms in aqueous systems is important not only for radiation damage but also in a broader context, making the reported results highly relevant to a wide audience. Therefore, I support the publication of this work in Nature Communications. However, there are several issues that the authors should address before I can recommend publication.

We thank the reviewer for their accurate summary, positive evaluation and for their thoughtful recommendations.

1) In my opinion, the main finding of this study is that proton transfer is not the sole mechanism responsible for the observed reduced ICD efficiency in liquid water. This point, however, is not emphasized enough in the manuscript. It is actually mentioned only briefly in the Discussion and Conclusions section. The authors state, "This discrepancy led us to hypothesize additional non-adiabatic energy relaxation mechanisms", yet no such mechanisms are discussed in the manuscript. The authors should elaborate on this point and propose a plausible mechanism. Moreover, in the Abstract, the proton transfer is still presented as the only explanation.

We are grateful to the reviewer for this opportunity to further emphasize the importance of the non-adiabatic relaxation mechanisms. In addition to the Discussion and Conclusions section, the non-adiabatic relaxation is also mentioned in the title and the abstract. However, we have now emphasizes the point also in the introduction.

2) In relation to the previous point. I find the ADC(2,2) results reported in Fig. 4 very important, as they

demonstrate that one cannot consider a single 2a1 state and potential, but rather a band of non-adiabatically coupled states spread over several eVs. Isn't this the key to understanding the complex non-adiabatic dynamics initiated by inner-valence ionization? It is not clear from the description of the computational scheme how the existence of the 2a1 band has been incorporated into the semi-classical simulations. The authors continue to talk about a single 2a1 potential (see, e.g., page 10).

The reviewer correctly identifies that the nonadiabatic dynamics in the 2a1 band underpin the additional relaxation mechanisms suggested by our observations and analysis. However, in the current simulations, the ADC(2,2) potentials are not explicitly used due to the unavailability of the necessary methodology, specifically the nonadiabatic couplings. Instead, our model accounts for the band character and nonadiabaticity stochastically through the energies shown in Fig. 5, derived from dynamical simulations, and phenomenologically through the parameter δ . The latter is now more explicitly linked to the nonadiabatic dynamics on page 11, where we have added:

The physics behind this parameter is the additional energy reduction due to the nonadiabatic decays of the singly ionized state in a dense manifold of electronic states.

3) When deriving the equation for ICD efficiency, the authors state on page 4 of the SI that the ICD process can occur only in the liquid phase, implicitly assuming that no dimers (or larger clusters) form in the vapor surrounding the jet. This assumption is not self-evident to me. Can the authors provide further arguments to support it?

Studies on the evaporation dynamics of liquid water jets in vacuum have explored whether the evaporating species are solely monomers or include clusters. Early molecular beam experiments demonstrated that for water microjets with diameters less than 10 μm , the evaporating molecules exhibit a Maxwellian velocity distribution corresponding to the jet's local temperature, indicating monomer evaporation under collision-free conditions:

<https://pubs.acs.org/doi/10.1021/acs.accounts.2c00739>

These results have been confirmed in quantitative analyses of liquid-water-jet photoelectron spectra that could be quantitatively modeled in terms of a liquid-bulk and a monomer contribution without evidence of clusters, which possess photoelectron spectra that are distinct from, both, the liquid bulk and the monomer:

<https://journals.aps.org/prl/abstract/10.1103/PhysRevLett.118.103402>

We have added this information to the SI. On page 5 of the SI, we have rewritten the sentence:

“Since the ICD process can only occur in the liquid phase” to read

“Given the experimentally confirmed negligible contribution of water clusters in the vapor surrounding the liquid jet under high-vacuum conditions \cite{Hartweg2017,Faubel2023}, we attribute the observed ICD process exclusively to the photoionisation of liquid water, and therefore ...”

4) In Fig. 6 (panels b and d), the authors report the ICD efficiency for the dimer as being well below 10%. Although this might be the case if ICD is initiated by ionizing the proton-donor molecule, the ionization of the proton acceptor should lead to decay with very high efficiency since no proton transfer is initiated and the decay channel remains open. Given that the probability of ionizing either molecule in the dimer is 50%, one would expect the overall efficiency in the dimer to exceed 50%. This situation, of course, changes in larger clusters where every molecule simultaneously acts as both proton donor and proton acceptor. The authors should either explain these results or remove the dimer from the figures reporting efficiency trends.

This is an important point that warrants discussion in the manuscript. Our current analysis assumes a scenario in which each water unit acts as a hydrogen-bond donor, which applies to all clusters except the dimer. Experimentally, even for systems with a mean cluster size of two, a significant fraction of larger clusters is typically present. The notably high ICD rate for the dimer - based on the 50% probability of ionizing the hydrogen-bond acceptor - is, however, debatable. The intrinsic ICD rate in the liquid phase is estimated to be around 50 fs, within which both charge transfer between the two water units and their dissociation could occur, potentially reducing the ICD probability. Nonetheless, these arguments remain speculative.

We have specifically highlighted the case of the dimer in our discussion, acknowledging that the water

dimer serves as a simplified model for small clusters—a model that allows for explicit computational treatment.

A few more minor issues:

1) In Fig. 2 (and 6) the authors report results from two different background subtraction methods. Although details are certainly not needed, what is meant by "lin.BG" and "shirley.BG" should somehow be pointed out in the caption.

We have added two sentences together with the references to read "lin.BG" and "shirley.BG" in the caption of fig.2:

"A simple linear background (lin.BG) model, consistent with the analysis in ref.\cite{Foerstel2013}. A shirley background model, which is widely used in x-ray photoelectron spectroscopy analysis \cite{Shirley1972}."

2) On page 7 of the manuscript, the authors discuss the breakdown of the molecular-orbital picture, giving rise to the population of a multitude of states by removing an electron from the 2a1 orbital. The authors should cite some of the original works on the subject.

We have added the new references 42 and 43:

<https://iopscience.iop.org/article/10.1088/0031-8949/21/3-4/040>

<https://www.sciencedirect.com/science/article/pii/S0009261424005256>

3) The authors should include the color bars in Fig S1.

We thank the reviewer for pointing this out. We have added the color bars in the revised Fig. S1.

4) First sentence of the last paragraph on page 16: "... the doubly-ionized states can decay into...". The authors likely mean that the singly-ionized states can decay to eleven doubly-ionized states.

We thank the reviewer for noticing this error that we have corrected.

5) The redundant letter "a" should be removed from panel e of Fig. S2.

We thank the reviewer for pointing out this typo in the figure and we have corrected it.

Reviewer #3 (Remarks to the Author):

The manuscript "ICD in liquid water competes with proton transfer and non-adiabatic relaxation", the authors report a coincidence photoelectron spectroscopy study of electronic and nuclear dynamics following ionization of liquid water with 63.5 eV photons. The experiments extend the work recently published in Ref. 31 by the experimental team to compare the dynamics of the heavy hydrogen atoms in D₂O with the previously-reported H₂O results. The data are interpreted with the aid of quantum chemistry electronic structure calculations, ab initio molecular dynamics simulations of water clusters, adapted to the electronic structure of liquid water with empirical ICD and nonadiabatic relaxation rates. The authors also pointed out the experimental challenges and the importance of studying ICD to understand many aspects of secondary or low-energy electrons relevant to modeling radiation damage. They claim from their experimental results and theoretical simulations that proton transfer and non-adiabatic relaxation are the main reasons for ICD, which contrasts with the ref [24]. Overall, the results reported in the manuscript are important incremental advances on recently published closely related work, specifically Ref. 31 and Slavicek et al. dx.doi.org/10.1021/ja5117588, J. Am. Chem. Soc. 2014, 136, 18170–18176).

We thank the reviewer for their summary. To avoid a potential misunderstanding, we would like to clarify that we are not claiming that proton transfer and non-adiabatic relaxations are the main reasons for ICD. On the contrary, these mechanisms suppress ICD because they compete with it.

The presence of low energy electrons measured in coincidence with inner-valence (2a₁) photoelectrons is clearly consistent with ICD. However, local autoionization processes, such as Auger-Meitner decay may also compete with ICD. Further analysis of local autoionization processes is needed to justify the assignment of the spectral region between the red lines in Fig. 1 b and d as "... dominated by 2a₁-photoelectron/ICD-electron pairs". To accurately determine the ICD efficiency, ICD must be isolated from all other ionization processes. How is local autoionization excluded? The authors must address this question before the manuscript will be suitable for publication.

We thank the reviewer for the opportunity to clarify this important aspect. The vertical double ionization potential (DIP) of a gas phase water molecule lies above 39eV [Truong et al, Chem. Phys. Lett. 474, 41 (2009), <https://doi.org/10.1016/j.cplett.2009.04.036>], which is the upper binding energy limit of the window for the photoelectron e₁ with kinetic energy 24.5eV to enter the ICD statistics (Fig 1). The double ionization of a water molecule observed in the above reference below the vertical DIP is due to the dissociative autoionization of H⁺-OH⁺ opening at H-OH distances above 2Å. The proton transfer in liquid is closely related to this mechanism.

Indeed, as identified in the new Tab. 1, for the O-H distances above 1.5Å the lowest dicationic states correspond to OH⁺...H₃O⁺ charge distribution. At these geometries, the electronic decay can thus be classified as autoionization (AI) of the inner-valence excited OH* fragment, resembling the previously identified *proton transfer-mediated Auger decay* in the case of core ionization, see Refs. [47,48]. In the new simulations performed for this revision, we have allowed for the reopening of the electronic decay channel and identified only up to 4.5% of the decay events to be associated with these AI-like transitions. The AI thus cannot be entirely ruled out, but our model predicts the contribution to be low. It should also be pointed out that the boundary between ICD and proton transfer-mediated AI is blurred as the lowest dicationic adiabatic states change their character along the proton transfer coordinate. In particular, for the dynamical trajectories staying above DIP along the whole proton transfer, the exact distinction is impossible in principle.

In the paper, we have added this discussion on page 12:

The majority of the emitted electrons originate from the ICD process. However, a detailed analysis reveals that a fraction of the electrons arises from highly stretched configurations, corresponding to OH*...H₃O⁺ structures. The autoionization of these states can be classified as a local Auger decay. This overall mechanism closely resembles the proton-transfer-mediated decay previously described for hydrogen-bonded systems following core-electron ionization [47,48]. The contribution of these decay events is 4.5% for the largest cluster considered, and for regular water, the contribution is smaller for smaller clusters and deuterated water.

and page 13:

The present study also suggests that a smaller fraction of the electronic decays can be classified as local proton transfer mediated decay. The final product would then be OH⁺ and H₃O.

In addition the above primary concern, the color scale should be added to Figure S1.

We have done this.

Reviewer #4 (Remarks to the Author):

We also thank reviewer #4 for their time and their valuable contributions to the peer review of this manuscript.

Reviewer #5 (Remarks to the Author):

This combined experimental and theory work establishes that the efficiency of intermolecular Coulombic decay (ICD) is below unity in both liquids, water and heavy water. With a relative efficiency of around 86% it is less likely to take place in water than in heavy water. The calculations suggest that ICD stands in competition with proton transfer between neighboring water molecules and non-adiabatic relaxation, which can both close the ICD channel.

The article is well structured and the narration is for the most part easy to follow (see some comments below); there are no language problems. The reference list is comprehensive. The figures are of good quality; however, some graphs would profit from some small changes (see comments below). In general, the topic of ICD is of high and broad interest to the research community.

We thank the reviewer for their accurate summary and for pointing out the broad relevance of ICD to the research community.

Yet, I am uncertain if this work fits the requirements of Nat. Commun., as, despite the advance, I do not see a real breakthrough in the problem. Nat. Comm. is supposed to be of interest for the general readership, and I imagine a general reader would want to know why something happened, not simply that there are experiments that agree with predictions from theory and vice versa. The authors find ICD electron energy distributions and efficiency trends that show agreement between experiment and theory and partly with previous cluster measurements. Using a fitting procedure, lifetimes and relaxation rates of ICD are found that are consistent with prior calculations on systems like neon and water clusters. The authors then infer that the 2a₁ one-hole state may initiate dissociation dynamics that is subject to non-adiabatic transitions. Yet, no further analysis, details or proof of this speculation is given here, which makes me think that, in the current state, this article would perhaps better fit a journal like, e.g., Phys. Rev. A or J. Phys. B.

We thank the reviewer for these thoughtful comments and the justified request to clarify the key breakthroughs of the present work.

The first key achievement of this work is the experimental measurement of the relative ICD efficiencies of two liquids through electron-electron coincidence spectroscopy using liquid microjets. This is a new experimental capability that opens the door to many interesting experiments, ranging from stationary measurements of ICD of different solvated molecules to time-resolved measurements that will track the intermediate states of the ICD process.

However, the key breakthrough in this work is that it really explains how ICD following inner-valence ionization of liquid water occurs and which physical effects matter. In this sense, our work does provide mechanistic insights into liquid-phase ICD, significantly advancing the state of the art. Importantly, our work shows the importance of the breakdown of the orbital approximation for ICD (Fig. 4), it shows the effect that solvation has on the potential-energy surfaces governing ICD (Fig. 5) and it explains how and, importantly, why the ICD of liquid water is not equal to unity (Fig. 6). This mechanistic insight is made possible through a multi-scale model that can explain not only the present liquid-phase results, but also the previous cluster results, providing a unified picture of ICD from dimers to bulk liquids. It is this new mechanistic understanding that we view as the key breakthrough of our work.

Besides this problem, I find the following deficiencies below. Please accept my comments, suggestions, and recommendations as efforts to improve the current draft:

P. 4, sentence “Nevertheless, the secondary-primary electron pairs are even more pronounced, which is caused by a higher electron-impact ionization cross section of D₂O [34].”

I fail to see that more low energy electron pairs are produced in Fig. 1d than in Fig. 2b.

We believe that the reviewer is referring to Fig. 1b where they write Fig. 2b and we thank the reviewer for inviting us to clarify this point.

In Fig.R1 below, the low-energy electron-pair distribution of liquid H₂O (black solid line, data taken from Fig. 1b) and D₂O (red solid line, data taken from Fig.1d) are plotted together for comparison. It is clearly seen that the secondary electron from liquid D₂O with very low kinetic energy is slightly higher than that from liquid H₂O. To better illustrate this point, we have added this figure to the supplementary

information as Figure. S2.

Fig. R1: Comparison of the low-energy electron-pairs distribution emitted from liquid H₂O (black solid line) and D₂O (red solid line), respectively. The spectra are normalised to the total electron-pair counts for comparison.

Accordingly, in the main text, we refer to this statement to Figure. S2 in SI:

“Nevertheless, the secondary-primary electron pairs are even more pronounced (see figure.S2 in Supplementary Information), which is caused by a higher electron-impact ionization cross section of D₂O [34].”

P. 4, sentence “The doubly charged final states, which are populated via either direct photo-double-ionization, single photoionization of outer-valence band followed by electron-impact ionization or the ICD process, ...”

Could autoionization also contribute to the electron pair signal?

Local autoionization can indeed contribute to the signal, but our model predicts the contribution to be low. See the detailed discussion in the response to Reviewer 4 above.

P. 4, sentence “The intensity of the electron pairs at the higher-binding-energy part is partly due to direct photo-double-ionization and mainly caused by secondary electrons induced from electron-impact ionization of liquid water by primary photoelectrons [31, 36].”

I am confused. Do you want to say that these electron pairs are generated via direct double ionization, but that both electrons suffer from loss in energy due to scattering? Please clarify.

We have rewritten this sentence to read: “A small fraction of the electron pairs in the higher-binding-energy region (>35 eV) is due to the creation of local double-hole states, whereas the highest-binding energy part is dominated by secondary electrons created through electron-impact ionization of liquid water by primary photoelectrons [31, 36].”

P. 4, sentence “For the two liquids, the ICD spectra display a peak around 1 eV, which is probably caused by the presence of an escape barrier on the order of 0.2 eV [36].”

The authors speculate that the slight depletion in the electron kinetic energy distribution below 1 eV, as shown in Fig. 1a, might be due to a presence of an escape barrier. Yet, could this be just because of the technical difficulties to measure low energy electrons below 1 eV, as stated on page 13 (Experimental Methods)? Perhaps the applied bias of +0.9V cannot fully compensate the loss in detection efficiency? Are there any calibration measurements of known processes that prove otherwise? If not, I recommend to mention this possible technical reason that could be responsible for the energy

distributions in Fig. 1.

The referee is absolutely correct in stating that the decreasing detection efficiency towards zero kinetic energy could cause a signal decrease. However, we can safely exclude this possibility because we (and others) have performed measurements with much larger biases of up to -20 V. The decrease of the photoelectron signal towards 0 eV (of nascent kinetic energy) is unchanged over the whole range bias voltages from 0 to -20 V. We are therefore confident that it is a real effect. Detailed modeling of this effect, allowing to trace it back to an escape barrier is given in Ref. 36 (Gadeyne et al., Chem. Sci. 2022).

P. 5, sentence "Figure 2 shows the ICD efficiency..." I suggest to change to "Figure 2e shows the ICD efficiency..."

We thank the reviewer for this suggestion, and we have made this change.

P. 5., sentence "The raw ICD efficiencies of H₂O cluster (black) and D₂O cluster (green) as well as their relative ratio (blue)..."

For the benefit of the reader, I suggest to write something like "For comparison, we also show the ICD efficiencies of H₂O (black) and D₂O (green) as well as their relative ratio (blue) for a cluster size of N ~72, extracted from Ref. [24], in Fig. 2e.

We thank the reviewer for this suggestion, and we have made this change.

P. 8, sentence "Around RO-H = 1.3 Angstrom, the lower part of the band drops below the lowest double-ionization potential, and the ICD channel partially closes. However, in contrast to the picture presented in Ref. [24] (cp. Fig. 3 therein), part of the inner-valence ionization band stays above the ICD threshold for all configurations."

+

P. 17, sentence "At each step, we check whether the kinetic energy, E_{kinetic} , is positive. If it is not, we terminate the trajectory."

In Ref. [24], the ICD channel reopens at O-H stretches > 1.8 Angstrom for the triplet state. While it is true that ICD is quenched quickly via the bond-elongation, it is possible that some events do not decay via internal conversion (IC) in the region of 1.2 to 1.8 Angstrom but undergo ICD at longer stretches. I think Fig. 4 in the draft shows this too, but it appears to me that this possibility has not been taken into account. Yet, intuitively, a reopening of the ICD channel would enhance the ICD yield for H₂O, because an O-H stretch would spend less time in the region between 1.2 and 1.8 Angstrom than an O-D stretch and, hence, would be less likely subject to IC. Please elaborate on this aspect.

Indeed, the reopening of the ICD channel is principally possible. We have ignored this possible scenario in our stochastic simulations: once we appeared in a configuration that does not allow ICD, the trajectory was terminated. We expect that (on average) the system will lose potential energy through non-adiabatic processes. Further, the decay rate for these stretched configurations might be different.

However, the reviewer made us rethink the concept. We have repeated the simulations with a modified algorithm that has allowed for reopening the decay channel. First, we were interested in how frequent these events are. For water dimer, these events made less than 1% of the total decay events - the exact number depends on the decay rate, the 1% value was found for 50 fs. The number reaches some 4.5% for the largest cluster (using the same parameters).

We have also performed preliminary calculations for the dimer using the Fano-ADC(2,2) method [44], suggesting that the decay rates will be approximately similar for the stretched configurations. To be specific, we have evaluated the decay widths for all the inner-valence ionized states of the dimer in the donor 2a₁ band shown in Fig 4 for the equilibrium and extreme proton-transfer geometries. We have estimated the effective decay width by an average weighted by the 2a₁ pole strength, resulting to an effective lifetime of 41fs for the equilibrium and 52fs for the PT geometry, consistent with the previously published estimates.

Note, however, that the decay for stretched configurations is technically a local Auger decay that was opened by the proton transfer. The processes could be classified as proton transfer mediated decay, see e.g. [dx.doi.org/10.1021/ja5117588](https://doi.org/10.1021/ja5117588).

We have redone all the simulations. There was no major change, yet the results appear to resemble the experimental ones even better than before. We now comment on this issue on page 12-13.

P. 11, sentence “This method necessitated an additional parameter, delta, which accounts for the systematic, time-dependent decrease in the energy of the singly-ionized state. The physics behind this parameter is the additional energy reduction due to the non-adiabatic decays of the singly ionized state in a dense environment.”

This is unclear to me. Please elaborate why an energy drop in the inner-valence-ionized state in a dense (neutral) environment is justified.

This was an unfortunate formulation. We had on mind that the 2a1 state appears in a dense manifold of electronic states (even as an isolated molecule). The “dense environment” is a clearly misleading term and we corrected for that.

SI, P. 4, sentence “Since the ICD process can only occur in the liquid phase, we have $\xi_{\text{ICD}}^{\text{gas}}=0$.”

I do not understand why ICD could not happen in the gas phase. Please elaborate.

ICD does not happen in the gas phase because the gas phase consists of monomers only, as discussed in response to question 3 or reviewer #2.

SI, P. 5, Fig. S2e:

I don't think the “a” in the graph has any meaning. Please delete.

We thank the reviewer for pointing out this typo, and we have deleted this in the renewed Fig.S2.

Fig. 1a and b + Fig. S1a and b:

Showing the error bars or stating something in the figure caption about the error would be appreciated.

We thank the reviewer for requesting this important clarification, please refer to the reply below.

Fig. 2a, b, c, and d + Fig. S2a, b, c, and d:

I recommend to compress the number of entries in the spectrum and instead show error bars. These figures also raise the question about the energy resolution of the measurement, which should likely be stated on page 13 (Experimental Methods). The bin size should not be much smaller than that,

We thank the reviewer for requesting this important clarification. Here, we discuss the energy resolution of the photoelectron spectrometer. Photoelectron detection was performed with a magnetic-bottle spectrometer, which was calibrated using the photoelectron spectrum generated by ionizing xenon and argon with different high-order harmonics (H11 to H25). Due to the parallelization mechanism of electron trajectories in the magnetic bottle, the photoelectron peaks exhibit an exponential tail towards decreasing kinetic energies. The spectral widths of the photoelectron peaks result from the convolution of the following contributions: the electronic responses of the MCP detector and the digitizer, the high-harmonic spectral width, the instrument response function of the magnetic-bottle spectrometer, and a possible natural line width.

The single-electron response of the MCP detector in the time domain can be described by a Gaussian function and has a full width at half-maximum of 0.9 ns. This temporal response can not be neglected when considering the absolute times of flight of the photoelectrons, which vary from 3 to 15 ns, when induced by the XUV harmonic orders ranging from H11 to H25. However, the digitizer's sampling rate is

1 gigasample/s, which fixes the minimum peak width to 1 ns. Consequently, the maximum achievable resolution is $0.15 \text{ eV}/26.87 \text{ eV} = 0.006$ for H25 and $0.009 \text{ eV}/3.72 \text{ eV} = 0.002$ for H11, respectively.

The XUV spectral width was experimentally determined by an XUV-photon spectrometer installed at the position of the slit selecting a single harmonic (see Fig. 7 in the experimental method). Harmonics entering the XUV spectrometer are dispersed by a grating with 600 lines/mm onto a MCP with a phosphor screen.

The instrument response function of the magnetic-bottle spectrometer can be expressed as the convolution of a Gaussian distribution and an exponential decay distribution:

$f(E)=A*\exp((d*d/(2*\tau*\tau))+((E/\tau)-E_0/\tau)*(1-\text{erf}((d*d-\tau*(E_0-E))/(\text{sqrt}(2)*d*\tau))))$, where A: amplitude; d: FWHM of Gaussian distribution; E_0 : center of energy; τ : decay constant;

Fig. R2: Energy resolution (ΔE) of the spectrometer as a function of kinetic energy.

From Fig.R2, one can clearly see that the energy resolution is kinetic energy-dependent. Nevertheless, the energy resolution shows a linear increasing behavior as the kinetic energy decreases.

Following the reviewer's suggestion, we compress the number of entries in the spectrum to align the energy resolution, the renewed figures are attached below as Fig. R3 (same as figure 2 in the resubmitted manuscript) and Fig. R4(same as figure S3 in the resubmitted supplementary material), respectively.

Fig. R3: Efficiency of ICD after photoionization of the $2a_1$ band in liquid water with different XUV photon energies.

Fig.R4 Illustration of the background subtraction in the experimental data used to determine the relative efficiency of ICD.

Fig. 4: Please label the various PECs.

We thank the referee for bringing out this point.

To facilitate the readability of figure 4 in the main text, we labeled the energy levels of singlet and triplet doubly outer-valence-ionized states (see Figure R3 below).

Fig. R3 Calculated potential-energy curves of singly- and doubly-ionized water dimer along the proton transfer coordinate. The intensity map shows pole strengths of the singly-ionized states in the inner-valence spectral region, manifesting a breakdown of the molecular-orbital picture. Solid and dashed red lines correspond to the few lowest-lying singlet and triplet doubly outer-valence-ionized states, respectively. For the states located around the equilibrium geometry, A and D stands for acceptor and donor, respectively. For the states located at the proton transfer geometry, A and D stands for H_3O^+ and OH^- , respectively.

To make a clear assignment of the configurations corresponding to their energy levels, we have created a new table contains all of the calculated dicationic states:

Table 1 Character of the dicationic states. For the equilibrium geometry, both the singlet and triplet states correspond to $\text{H}_2\text{O}^+ - \text{H}_2\text{O}^+$. For the proton transfer geometry, the lowest three singlet and triplet states are $\text{OH}^+ - \text{OH}_3^+$, the highest, fourth state is $\text{OH} - \text{OH}_3^{2+}$.

Geometry	Energy (eV)	Singlet	Triplet
Equilibrium (A: acceptor; D: donor)	33.2	$(\text{HOMO})^D(\text{HOMO})^A$	
	31.0	$(\text{HOMO}-1)^D(\text{HOMO})^A$	
	30.8	$(\text{HOMO})^D(\text{HOMO}-1)^A$	
	28.7	$(\text{HOMO}-1)^D(\text{HOMO}-1)^A$	
Proton Transfer (A: H_3O^+ ; D: OH^-)	33.2	$(\text{HOMO})^A(\text{HOMO})^D$	
	32.2	$(\text{HOMO}-1)^D(\text{HOMO}-1)^D$	
	30.9	$(\text{HOMO}-1)^D(\text{HOMO})^D$	
	30.9	$(\text{HOMO})^D(\text{HOMO})^D$	
	33.2		$(\text{HOMO})^A(\text{HOMO})^D$
	32.6		$(\text{HOMO}-1)^D(\text{HOMO}-2)^D$
	32.3		$(\text{HOMO})^D(\text{HOMO}-2)^D$
28.8		$(\text{HOMO})^D(\text{HOMO}-1)^D$	

Fig. 5 + Fig. S3: See comments to Fig. 2a, b, c, and d. Moreover, there seems to be a sinusoidal substructure in the singly ionized 2a1 states. Is this an artefact of the calculation or something real?

We did not pay attention to these fine features. It is most likely merely an artifact resulting from variations in the initial OH bond distances at the start of the OH bond scan. These differences introduce a shift in the curves, which produces a sinusoidal pattern.

We thank the reviewers and the editor for their reviews and thoughtful comments. Below, we reproduce their statements in blue, provide our replies in black and the changes made to the manuscript in green.

Referees' comments:

Referee #1 (Remarks to the Author):

The revised manuscript addresses all my suggestions. I am glad to recommend publication of this manuscript in nature communications

We thank the reviewer for their time that they have dedicated to the review of our manuscript and for recommending publication.

Reviewer #2 (Remarks to the Author):

The authors have thoroughly addressed all of my previous criticisms and recommendations, as well as those raised by the other reviewers. They have revised the manuscript by extending the discussion and clarifying several key aspects of their approach and analysis. In addition, they have performed new simulations and refined their model, which now shows even better agreement with the experimental data.

As I already stated in my initial report, I believe the manuscript presents results of general interest and contains important new findings that merit publication in Nature Communications. I therefore recommend the paper for publication.

We are grateful to the reviewer for all of their constructive criticism and thoughtful recommendations, as well as for recommending publication of our manuscript.

I would like to point out one minor issue: in the revised abstract, the abbreviation ICD is defined twice, which is obviously unnecessary.

We have removed this repetition.

Reviewer #3 (Remarks to the Author):

In the revised manuscript, revised supplementary information, and responses to the referee reports, the authors have thoroughly addressed each of my comments and concerns, as well as those of the other referees. I recommend the manuscript publication in Nature Communications without further revision.

We thank the reviewer for their dedicated work on the review of this manuscript, which has substantially contributed to its improvement. We are grateful to the reviewer for recommending publication.

Reviewer #4 (Remarks to the Author):

We greatly appreciate the participation and the valuable contributions of the reviewer and for supporting the publication of our manuscript.

Reviewer #5 (Remarks to the Author):

Overall, after not too many changes, the draft has somewhat improved.

We acknowledge the reviewer's remarks and appreciate the recognition of the study's experimental innovation and modeling framework. Our goal was to bridge liquid and cluster observations, and we believe the approach provides a useful basis for further exploration of ICD in complex environments.

I agree that the paper introduces a novel experimental technique to measure the relative efficiencies of Interatomic Coulombic Decay (ICD) in liquids, using electron-electron coincidence spectroscopy with liquid microjets. I also concur, that this work highlights the breakdown of the orbital approximation and the effects of solvation on potential-energy surfaces. It is commendable, that the study uses a multi-scale model that unifies results across liquid-phase and cluster systems. Yet, the reasons why the ICD efficiency in liquid water is not unity is still elusive. It appears Reviewer 2 and I share the same concerns in terms of delivering this breakthrough. Given the title, introduction, and overall aspiration of this paper, the hypothesized non-adiabatic energy relaxation mechanisms are (still) not well discussed (just mentioning the word "non-adiabatic" more frequently in the paper is not really the improvement I was after).

We understand the reviewer's concern. The delta parameter was introduced to represent an additional energy loss during femtosecond dynamics, which we attribute to (admittedly unresolved) non-adiabatic processes. Without this parameter, we were not able to reproduce the reduced efficiency, but we also did not obtain reasonable distributions of kinetic energies. While we cannot offer a detailed mechanism at ab initio level, the parameter reflects systematic deviations that align with experimental trends.

The authors now clarify that these mechanisms are represented in their model by "...an additional parameter, δ , which accounts for the systematic, time-dependent decrease in the energy of the singly-ionized state. The physics behind this parameter is the additional energy reduction due to the non-adiabatic decays of the singly ionized state in a dense manifold of electronic states." This is helpful, yet, insights into the non-adiabatic transitions from the satellite states that formed during the creation of the 2a1 vacancy and which are sketched in Fig. 3 to the calculated PECs in Fig. 4 are lacking. From the response to Reviewer 2, I now understand that this is not possible, because "the ADC(2,2) potentials are not explicitly used due to the unavailability of the necessary methodology, specifically the nonadiabatic couplings." I think this will disappoint the general readership, as it leaves them with an answered problem.

As noted, we cannot currently compute non-adiabatic couplings within the ADC(2,2) framework for liquids. The initially excited inner valence state (in the liquid state) is embedded in an environment with a high density of states, with most of them negligibly coupled to the original state. This makes it virtually impossible to explicitly model the non-adiabatic dynamics at the ab initio level, and we need to describe the process phenomenologically with the delta parameter. We recognize this limitation and **now explicitly state in the manuscript that the mechanistic details remain inaccessible due to methodological constraints.**

If the draft goes forward to publication in Nat. Commun., which in my view it still does not really qualify for (see above), at a minimum the authors should clue in the general reader about this deficiency and dedicate a paragraph to explaining the current limitations, in order to put the speculations about non-adiabatic transitions in perspective and state the remaining open questions.

We have added a paragraph to the conclusion addressing this limitation. It explains the role of delta, the absence of direct simulation of non-adiabatic couplings, and the open questions that remain. While speculative elements are unavoidable at this stage, we believe they are clearly identified as such.

Some additional comments and suggestions:

- The new Fig. S2 and Table I are very helpful.

- I recommend adding a sentence or two on the answer about the decreasing detection efficiency towards zero kinetic energy in Section S.1, stating that this poses no problem for the measurement and the presented results, in order to put the reader at ease.

We greatly appreciate the valuable suggestion from the reviewer, and we have added one sentence in Supplementary Note 1 to clarify the role of negative bias potential on the liquid jet.

- Fig. 2 and S3 have improved. Yet, statistical error bars are still missing. If the authors don't want to explicitly show them, a comment on the uncertainty in the captions would also work. I also recommend to mention the approximate average energy resolution of the spectrometer (shown in Fig. R2) as ~55:1 in, e.g., the Experimental Methods section.

We thank the reviewer for the valuable comments and made corresponding revisions.

We have added one sentence in the captions of Fig.2 and S3 to clarify the uncertainty. We also added one sentence to mention the energy resolution of the spectrometer in the Experimental Methods section.

- Fig. 5 is still hard to read because of the scattered data points. If this cannot be smoothed out without compromising the calculated results, maybe at least mention that the oscillations blurring the data are an artifact resulting from the variations in the initial OH bond distances.

We thank the reviewer for this suggestion for improving the readability of this manuscript.

We have added one sentence in the caption of Fig. 5 (same as Fig. S4) to clarify the origin of the oscillating features due to the artifact during the computing process.